# Zeroth-Order Optimization Meets Human Feedback: Provable Learning via Ranking Oracles

**Zhiwei Tang**[1,3*]**, Dmitry Rybin**[2]**, Tsung-Hui Chang**[1,3]
School of Science and Engineering[1], School of Data Science[2]
The Chinese University of Hong Kong, Shenzhen, China
Shenzhen Research Institute of Big Data[3], Shenzhen, China
{zhiweitang1,dmitryrybin}@link.cuhk.edu.cn
changtsunghui@cuhk.edu.cn

## Abstract

In this study, we delve into an emerging optimization challenge involving a black-box objective function that can only be gauged via a ranking oracle—a situation frequently encountered in real-world scenarios, especially when the function is evaluated by human judges. Such challenge is inspired from Reinforcement Learning with Human Feedback (RLHF), an approach recently employed to enhance the performance of Large Language Models (LLMs) using human guidance (Ouyang et al., 2022; Liu et al., 2023; OpenAI, 2022; Bai et al., 2022). We introduce ZO-RankSGD, an innovative zeroth-order optimization algorithm designed to tackle this optimization problem, accompanied by theoretical assurances. Our algorithm utilizes a novel rank-based random estimator to determine the descent direction and guarantees convergence to a stationary point. Moreover, ZO-RankSGD is readily applicable to policy optimization problems in Reinforcement Learning (RL), particularly when only ranking oracles for the episode reward are available. Last but not least, we demonstrate the effectiveness of ZO-RankSGD in a novel application: improving the quality of images generated by a diffusion generative model with human ranking feedback. Throughout experiments, we found that ZO-RankSGD can significantly enhance the detail of generated images with only a few rounds of human feedback. Overall, our work advances the field of zeroth-order optimization by addressing the problem of optimizing functions with only ranking feedback, and offers a new and effective approach for aligning Artificial Intelligence (AI) with human intentions.

## 1 Introduction

Ranking data is an omnipresent feature of the internet, appearing on a variety of platforms and applications, such as search engines, social media feeds, online marketplaces, and review sites. It plays a crucial role in how we navigate and make sense of the vast amount of information available online. Moreover, ranking information has a unique appeal to humans, as it enables them to express their personal preferences in a straightforward and intuitive way (Ouyang et al., 2022; Liu et al., 2023; OpenAI, 2022; Bai et al., 2022). The significance of ranking data becomes even more apparent when some objective functions are evaluated through human beings, which is becoming increasingly common in various applications. Assigning an exact score or rating can often require a significant amount of cognitive burden or domain knowledge, making it impractical for human evaluators to provide precise feedback. In contrast, a ranking-based approach can be more natural and straightforward, allowing human evaluators to express their preferences and judgments with ease (Keeney & Raiffa, 1993). In this context, our paper makes the first attempt to study an important optimization problem where the objective function can only be accessed via a ranking oracle.

**Problem formulation.** With an objective function $f : \mathbb{R}^d \to \mathbb{R}$, we focus on the optimization problem $\min_{x \in \mathbb{R}^d} f(x)$, where $f$ is a black-box function, and we can only query it via a ranking oracle that can sort every input based on the values of $f$. In this work, we focus on a particular

---

*Correspondence to: Zhiwei Tang zhiweitang1@link.cuhk.edu.cn

family of ranking oracles where only the sorted indexes of top elements are returned. Such oracles are acknowledged to be natural for human decision-making (Keeney & Raiffa, 1993). We formally define this kind of oracle as follows:

**Definition 1** ($(m, k)$-ranking oracle). *Given a function $f : \mathbb{R}^d \to \mathbb{R}$ and $m$ points $x_1, ..., x_m$ to query, an $(m, k)$ ranking oracle $O_f^{(m,k)}$ returns $k$ smallest points sorted in their order. For example, if $O_f^{(m,k)}(x_1, ..., x_m) = (i_1, ..., i_k)$, then*

$$f(x_{i_1}) \leq f(x_{i_2}) \leq ... \leq f(x_{i_k}) \leq \min_{j \notin \{i_1, ..., i_k\}} f(x_j).$$

**Applications.** The optimization problem $\min_{x \in \mathbb{R}^d} f(x)$ with an $(m, k)$-ranking oracle is a common feature in many real-world applications, especially when the objective function $f$ is evaluated by human judges. One prominent inspiration for this type of problem is the growing field of Reinforcement Learning with Human Feedback (RLHF) (Ouyang et al., 2022; Liu et al., 2023; OpenAI, 2022; Bai et al., 2022), where human evaluators are asked to rank the outputs of large AI models according to their personal preferences, with an aim to improve the generation quality of these models. Inspired by these works, in Section 4, we propose a similar application in which human feedback is used to enhance the quality of images generated by Stable Diffusion (Rombach et al., 2022), a text-to-image generative model. An overview of this application is demonstrated in Figure 1. Beyond human feedback, ranking oracles have the potential to be useful in many other applications. For instance, in cases where the exact episode reward in reinforcement learning, or the precise values of the objective function $f$ must remain private, ranking data may provide a more secure and confidential option for data sharing and analysis. This is particularly relevant in sensitive domains, such as healthcare or finance, where the exact value of personal information must be protected.

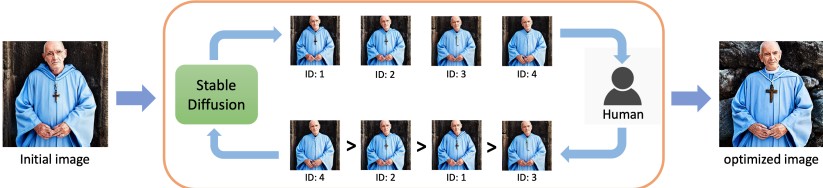

Figure 1: Application of our proposed algorithm on enhancing the quality of images generated from Stable Diffusion with human ranking feedback. At each iteration of this human-in-the-loop optimization, we use Stable Diffusion to generate multiple images by perturbing the latent embedding with random noise, which are then ranked by humans based on their quality. After that, the ranking information is leveraged to update the latent embedding.

## 1.1 Related works

**Zeroth-Order Optimization.** Zeroth-order optimization has been rigorously explored in the optimization literature over several decades (Nelder & Mead, 1965; Frazier, 2018; Golovin et al., 2019; Nesterov & Spokoiny, 2017). Despite this, most existing works make a significant assumption that the value of the objective function is directly accessible—an assumption ill-suited for our context, where only ranking data of the function value is available. Existing heuristic algorithms like CMA-ES (Loshchilov & Hutter, 2016), which exclusively rely on ranking information, often lack theoretical guarantees and may underperform in real-world scenarios. A notable exception is the recent study by (Cai et al., 2022), which investigates a setting where a pairwise comparison oracle of the objective function is available. This comparison oracle is indeed a $(2, 1)$-ranking oracle, making it a special case within our work's scope. (Cai et al., 2022) attempts to uncover the gradient of the objective function using the 1-bit compressive sensing method. Beyond (Cai et al., 2022), (Yue & Joachims, 2009; Ding & Zhou, 2018; Kumagai, 2017) also study the use of comparison oracle, but in the context of online bandit optimization. One major problem in all these existing works on comparison oracles is that the underlying objective is confined to be convex/strongly-convex, which is particularly unrealistic in some applications involving human preference. Our work, in contrast, contemplates a more general $(m, k)$-ranking oracle and focuses primarily on non-convex functions. Rather than relying on compressive sensing techniques, our work introduces a novel theoretical analysis capable of characterizing the expected convergence behavior of our proposed algorithm.

**Bayesian Optimization with Comparison Oracles.** Another relevant topic is Bayesian optimization using pairwise comparison oracles, as demonstrated in (Astudillo & Frazier, 2020) and (Lin

et al., 2022). Compared to the approach studied in this work, their approaches have two key issues. Firstly, unlike gradient-based algorithms, these works lack strong theoretical guarantees for optimization. Moreover, similar to the CMA-ES algorithm (Loshchilov & Hutter, 2016), Bayesian Optimization faces scalability issues and struggles with high-dimensional optimization, which is not a problem for gradient-based algorithms, as shown in (Duchi et al., 2015).

**Reinforcement Learning with Human Feedback (RLHF).** The general approach in existing RLHF procedures involves collecting human ranking data to train a reward model, which is then used to finetune a pre-trained model with policy gradients (Ouyang et al., 2022; Liu et al., 2023; OpenAI, 2022; Bai et al., 2022). In this work, we explore an alternative setting that fuses reinforcement learning with ranking feedback, where ranking occurs online and is based on the total reward of the entire episode. Our proposed zeroth-order algorithm can be directly employed to optimize the policy within this context.

**Contributions in this work.** Our main contributions are summarized as follows:

(1) **First rank-based zeroth-order optimization algorithm with theoretical guarantee.** We present a novel method for optimizing objective functions via their ranking oracles. Our proposed algorithm ZO-RankSGD is based on a new rank-based stochastic estimator for descent direction and is proven to converge to a stationary point. Additionally, we provide a rigorous analysis of how various ranking oracles can impact the convergence rate by employing a novel variance analysis. Last but not least, ZO-RankSGD is also directly applicable to the policy search problem in reinforcement learning with only a ranking oracle of the episode reward available.

(2) **A new method for using human feedback to guide AI models.** ZO-RankSGD offers a fresh and effective strategy for aligning human objectives with AI systems. We demonstrate its utility by applying our algorithm to a novel task: enhancing the quality of images generated by Stable Diffusion with human ranking feedback. We anticipate that our approach will stimulate further exploration of such applications in the field of AI alignment.

**Notations.** For any $x \in \mathbb{R}$, we define the sign operator as $\text{Sign}(x) = 1$ if $x \geq 0$ and $-1$ otherwise, and extend it to vectors by applying it element-wise. For a $d$-dimensional vector $x$, we denote the $d$-dimensional standard Gaussian distribution by $\mathcal{N}(0, I_d)$. The notation $|\mathcal{S}|$ refers to the number of elements in the set $\mathcal{S}$.

**Paper organization.** The rest of this paper is structured as follows: Section 2 introduces how to estimate descent direction based on ranking information, with a theoretical analysis of how different ranking oracles relate to the variance of the estimated direction. Built on the foundations in Section 2, Section 3 presents the main algorithm, ZO-RankSGD, along with the corresponding convergence analysis. In Section 4, we demonstrate the effectiveness of ZO-RankSGD through various experiments, ranging from synthetic data to real-world applications. Finally, Section 5 concludes the paper by summarizing our findings and suggesting future research directions.

## 2 FINDING DESCENT DIRECTION FROM THE RANKING INFORMATION

**Assumption 1.** *Throughout this paper, we have these assumptions on the function $f$: (1) $f$ is twice continuously differentiable. (2) $f$ is $L$-smooth, meaning that $\|\nabla^2 f(x)\| \leq L$. (3) $f$ is lower bounded by a value $f^*$, that is, $f(x) \geq f^*$ for all $x$.*

### 2.1 A COMPARISON-BASED ESTIMATOR FOR DESCENT DIRECTION

In contrast to the prior work (Cai et al., 2022), which relies on one-bit compressive sensing to recover the gradient, we propose a simple yet effective estimator for descent direction without requiring solving any compressive sensing problem. Given an objective function $f$ and a point $x$, we estimate the descent direction of $f$ using two independent Gaussian random vectors $\xi_1$ and $\xi_2$ as follows:

$$\hat{g}(x) = S_f(x, \xi_1, \xi_2, \mu)(\xi_1 - \xi_2), \tag{1}$$

where $\mu > 0$ is a constant, and $S_f(x, \xi_1, \xi_2, \mu) : \mathbb{R}^d \times \mathbb{R}^d \times \mathbb{R}^d \times \mathbb{R}_+ \to \{1, -1\}$ is defined as:

$$S_f(x, \xi_1, \xi_2, \mu) \overset{\text{def.}}{=} \text{Sign}\left((f(x + \mu\xi_1) - f(x + \mu\xi_2))\right). \tag{2}$$

We prove in Lemma 1, which is one of the most important technical tools in this work, that $\hat{g}(x)$ is an effective estimator for descent direction.

**Lemma 1.** *For any $x \in \mathbb{R}^d$, we have*

$$\langle \nabla f(x), \mathbb{E}[\hat{g}(x)] \rangle \geq \|\nabla f(x)\| - C_d \mu L, \tag{3}$$

*where $C_d \geq 0$ is some constant that only depends on $d$.*

Denote $\gamma > 0$ as the step size. With the $L$-smoothness of $f$ and Lemma 1, we can show that

$$\mathbb{E}_{\xi_1,\xi_2} [f(x - \gamma \hat{g}(x))] - f(x) \leq -\gamma \langle \nabla f(x), \mathbb{E}[\hat{g}(x)] \rangle + \frac{\gamma^2 L}{2} E\left[\|\hat{g}(x)\|^2\right]$$

$$\leq -\gamma \|\nabla f(x)\| + \gamma C_d \mu L + \gamma^2 L d, \tag{4}$$

where we note that $\mathbb{E}[\|\hat{g}(x)\|^2] = \mathbb{E}[\|\xi_1 - \xi_2\|^2] = 2d$. Therefore, whenever $\|\nabla f(x)\| \neq 0$, the value $\mathbb{E}_{\xi_1,\xi_2}[f(x - \gamma \hat{g}(x))]$ would be strictly smaller than $f(x)$ with sufficiently small $\gamma$ and $\mu$. More importantly, unlike the comparison-based gradient estimator proposed in (Cai et al., 2022), our estimator (1) can be directly incorporated with ranking oracles, as we will see in the next section.

## 2.2 FROM RANKING INFORMATION TO PAIRWISE COMPARISON

We first observe that ranking information can be translated into pairwise comparisons. For instance, knowing that $x_1$ is the best among $x_1, x_2, x_3$ can be represented using two pairwise comparisons: $x_1$ *is better than* $x_2$ and $x_1$ *is better than* $x_3$. Therefore, we propose to represent the input and output of $(m,k)$-ranking oracles as a directed acyclic graph (DAG), $\mathcal{G} = (\mathcal{N}, \mathcal{E})$, where the node set $\mathcal{N} = \{1, \ldots, m\}$ and the directed edge set $\mathcal{E} = \{(i,j) \mid f(x_i) < f(x_j)\}$. An example of such a DAG is shown in Figure 2. Given access to an $(m,k)$-ranking oracle $O_f^{(m,k)}$ and a starting point $x$, we query $O_f^{(m,k)}$ with the inputs $x_i = x + \mu \xi_i, \xi_i \sim \mathcal{N}(0, I_d)$, for $i = 1, \ldots, m$. With the graph $\mathcal{G}$ constructed from the ranking information of $O_f^{(m,k)}$, we propose the following rank-based gradient estimator:

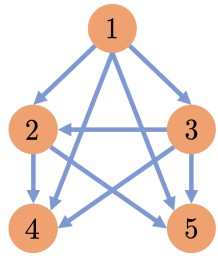

Figure 2: The corresponding DAG for the ranking result $O_f^{(5,3)}(x_1, x_2, x_3, x_4, x_5) = (1, 3, 2)$.

$$\tilde{g}(x) = \frac{1}{|\mathcal{E}|} \sum_{(i,j) \in \mathcal{E}} \frac{x_j - x_i}{\mu} = \frac{1}{|\mathcal{E}|} \sum_{(i,j) \in \mathcal{E}} (\xi_j - \xi_i). \tag{5}$$

**Remark 1.** *Notice that* (5) *can be simply expressed as a linearly weighted combination of $\xi_1, \ldots, \xi_m$. We provide the specific form in Appendix A.*

We note that (1) is a special case of (5) with $m = 2$ and $k = 1$, and it can be easily shown that $\mathbb{E}[\tilde{g}(x)] = \mathbb{E}[\hat{g}(x)]$ and $\mathbb{E}[\|\tilde{g}(x)\|^2] \leq \mathbb{E}[\|\hat{g}(x)\|^2]$, indicating that the benefit of using ranking information over a single comparison is a reduced variance of the gradient estimator. However, to determine the extent of variance reduction, we must examine the graph topology of $\mathcal{G}$.

**Graph topology of $\mathcal{G}$.** The construction of the DAG $\mathcal{G}$ described above reveals that the graph topology of $\mathcal{G}$ is uniquely determined by $m$ and $k$. There are two important statistics in this graph topology. The first one is the number of edges $|\mathcal{E}|$, which is related to the number of pairwise comparisons, extracted from the ranking result. In the precedent work (Cai et al., 2022), the number of pairwise comparisons can be used to determine the variance of the gradient estimator. However, this is insufficient for our case, as the pairwise comparisons in (5) are not independent. Therefore, we require the second statistic of the DAG, which is the number of neighboring edge pairs in $\mathcal{E}$. We define a neighboring edge pair as a pair of edges that share the same node. For instance, in Figure 2, one neighboring edge pair is $(x_1, x_3)$ and $(x_1, x_2)$. We denote this number as $N(\mathcal{E})$ and define it formally as $N(\mathcal{E}) \stackrel{\text{def.}}{=} |\{((i,j),(i',j)) \in \bar{\mathcal{E}} \times \bar{\mathcal{E}} \mid i \neq i'\}|$, where $\bar{\mathcal{E}}$ is the undirected copy of $\mathcal{E}$, i.e, $(i,j) \in \bar{\mathcal{E}}$ if and if only $(j,i)$ in $\mathcal{E}$ or $(i,j)$ in $\mathcal{E}$. As mentioned, the graph topology of $\mathcal{G}$ is determined by $m$ and $k$. Therefore, we can analytically compute $|\mathcal{E}|$ and $N(\mathcal{E})$ using $m$ and $k$. We state these calculations in the following lemma:

**Lemma 2.** *Let $\mathcal{G} = (\mathcal{N}, \mathcal{E})$ be the DAG constructed from the ranking information of $O_f^{(m,k)}$. Then,*

$$|\mathcal{E}| = km - (k^2 + k)/2, \tag{6}$$

$$N(\mathcal{E}) = m^2 k + mk^2 - k^3 + k^2 - 4mk + 2k. \tag{7}$$

**Variance analysis of** (5) **based on the graph topology.** To analyze the variance of the estimator (5), we introduce two important metrics $M_1(f, \mu)$ and $M_2(f, \mu)$ on the function $f$.

**Definition 2.**

$$M_1(f, \mu) \overset{def.}{=} \max_x \left\| \mathbb{E}_{\xi_1, \xi_2} \left[ S_f(x, \xi_1, \xi_2, \mu)(\xi_1 - \xi_2) \right] \right\|^2, \tag{8}$$

$$M_2(f, \mu) \overset{def.}{=} \max_x \mathbb{E}_{\xi_1, \xi_2, \xi_3} \left[ S_f(x, \xi_1, \xi_2, \mu) S_f(x, \xi_1, \xi_3, \mu) \langle \xi_1 - \xi_2, \xi_1 - \xi_3 \rangle \right], \tag{9}$$

*where $\xi_1$, $\xi_2$ and $\xi_3$ are three independent random vectors drawn from $\mathcal{N}(0, I_d)$.*

We also provide some useful upper bounds on $M_1(f, \mu)$ and $M_2(f, \mu)$ in Lemma 3, which help to understand the scale of these two quantities.

**Lemma 3.** *For any function $f$ and $\mu > 0$, we have $M_1(f, \mu) \le 2d$, $M_2(f, \mu) \le 2d$. Moreover, if $f$ satisfies that $\nabla^2 f(x) = cI_d$ where $c \in \mathbb{R}$ is some constant, we have $M_1(f, \mu) \le 32/\pi$.*

With $M_1(f, \mu)$ and $M_2(f, \mu)$, we can bound the second order moment of (5) as shown in Lemma 4.

**Lemma 4.** *For any $x \in \mathbb{R}^d$, we have*

$$\mathbb{E}[\|\tilde{g}(x)\|^2] \le \frac{2d}{|\mathcal{E}|} + \frac{N(\mathcal{E})}{|\mathcal{E}|^2} M_2(f, \mu) + M_1(f, \mu). \tag{10}$$

**Discussion on Lemma 4.** With Lemma 2 and Lemma 3, we observe that the first variance term in (10), namely, $\frac{2d}{|\mathcal{E}|}$, is $\mathcal{O}(\frac{1}{km})$, and thus vanishes as $m \to \infty$. In contrast, the second variance term $\frac{N(\mathcal{E})}{|\mathcal{E}|^2} M_2(f, \mu)$ does not disappear as $m$ grows, because

$$\lim_{m \to \infty} \frac{N(\mathcal{E})}{|\mathcal{E}|^2} = \lim_{m \to \infty} \frac{m^2 k + mk^2 - k^3 + k^2 - 4mk + 2k}{\left(km - (k^2 + k)/2\right)^2} = \frac{1}{k}, \tag{11}$$

and thus only vanishes when both $k$ and $m$ tend to infinity. However, there is a non-diminishing term $M_1(f, \mu)$ remaining in (10). Fortunately, as shown in Lemma 3, $M_1(f, \mu)$ is smaller than $2d$ and can be bounded by a dimension-independent constant for a certain family of quadratic functions. Finally, it is worth noting that our approach for the variance analysis can be directly extended to any ranking oracles beyond the $(m, k)$-ranking oracle.

# 3 ZO-RANKSGD: ZEROTH-ORDER RANK-BASED STOCHASTIC GRADIENT DESCENT

With all of our findings in Sections 2, now we are ready to introduce our proposed algorithm, ZO-RankSGD. The pseudocode for ZO-RankSGD is outlined in Algorithm 1.

---

**Algorithm 1** ZO-RankSGD

**Require:** Initial point $x_0$, stepsize $\eta$, number of iterations $T$, smoothing parameter $\mu$, $(m, k)$-ranking oracle $O_f^{(m,k)}$.

1: **for** $t = 1$ to $T$ **do**
2:      Sample $m$ i.i.d. random vectors $\{\xi_{(t,1)}, \cdots, \xi_{(t,m)}\}$ from $N(0, I_d)$.
3:      Query the $(m, k)$-ranking oracle $O_f^{(m,k)}$ with input $\{x_{t-1} + \mu\xi_{(t,1)}, \cdots, x_{t-1} + \mu\xi_{(t,m)}\}$, and construct the corresponding DAG $\mathcal{G} = (\mathcal{N}, \mathcal{E})$ as described in Section 2.2.
4:      Compute the gradient estimator using: $g_t = \frac{1}{|\mathcal{E}|} \sum_{(i,j) \in \mathcal{E}} (\xi_{(t,j)} - \xi_{(t,i)})$
5:      $x_t = x_{t-1} - \eta g_t$.
6: **end for**

---

## 3.1 THEORETICAL GUARANTEE OF ZO-RANKSGD

Now we present the convergence result of Algorithm 1 in the following Theorem 1.

**Theorem 1.** *For any $\eta > 0$, $\mu > 0$, $T \in \mathbb{N}$, after running Algorithm 1 for $T$ iterations, we have:*

$$\mathbb{E}\left[ \min_{t \in \{1, \ldots, T\}} \|\nabla f(x_{t-1})\| \right] \le \frac{f(x_0) - f^*}{\eta T} + C_d \mu L + \frac{\eta L}{2} \left( \frac{2d}{|\mathcal{E}|} + \frac{N(\mathcal{E})}{|\mathcal{E}|^2} M_2(f, \mu) + M_1(f, \mu) \right), \tag{12}$$

*where $C_d$ is some constant that only depends on $d$.*

**Corollary 1.** *By taking $\eta = \sqrt{\frac{1}{dT}}$ and $\mu = \sqrt{\frac{d}{C_d^2 T}}$ in Theorem 1, we have*

$$\mathbb{E}\left[\min_{t \in \{1,...,T\}} \|\nabla f(x_{t-1})\|\right] = \mathcal{O}\left(\sqrt{\frac{d}{T}}\right). \tag{13}$$

**Effect of $m$ and $k$ on the convergence speed of Algorithm 1.** As we have discussed in Section 2.2, $m$ and $k$ affect the convergence speed through the variance of the gradient estimator. Specifically, in the upper bound of (12), we have $\frac{2d}{|\mathcal{E}|} + \frac{N(\mathcal{E})}{|\mathcal{E}|^2} M_2(f, \mu) = \mathcal{O}\left(\frac{d}{km} + \frac{d}{k}\right)$.

**How to choose $\mu$ in Algorithm 1.** As we can see from (12), a smaller $\mu$ generally leads to a tighter bound as the estimated gradient aligns better with the true gradient. However, practical implementation demands striking a balance, as an excessively small $\mu$ might result in perturbed instances that are extremely similar to humans, making the ranking decision of them challenging. Therefore, a practical rule for choosing $\mu$ is: While we should try to minimize $\mu$, it should remain within the range of human discriminability.

### 3.2 LINE SEARCH VIA RANKING ORACLE

In this section, we discuss two potential issues that may arise when implementing Algorithm 1. Firstly, it can be cumbersome to manually tune the step size $\eta$ required for each iteration. Secondly, it may be challenging for users to know whether the objective function is decreasing in each iteration as the function values are not accessible. In order to address these challenges, we propose a simple and effective line search method that leverages the $(l, 1)$-ranking oracle to determine the optimal step size for each iteration. The method involves querying the oracle with a set of inputs $\{x_{t-1}, x_{t-1} - \eta \gamma g_t, ..., x_{t-1} - \eta \gamma^{l-1} g_t\}$, where $\gamma \in (0, 1)$ represents a scaling factor that controls the rate of step size reduction. By monitoring whether or not $x_t$ is equal to $x_{t-1}$, users can observe the progress of Algorithm 1, while simultaneously selecting a suitable step size to achieve the best results. It is worth noting that this line search technique is not unique to Algorithm 1 and can be applied to any gradient-based optimization algorithm, including those in (Nesterov & Spokoiny, 2017; Cai et al., 2022). To reflect this, we present the proposed line search method as Algorithm 2, under the assumption that the gradient estimator $g_t$ has already been computed.

---

**Algorithm 2** Line search strategy for gradient-based optimization algorithms

---

**Require:** Initial point $x_0$, stepsize $\eta$, number of iterations $T$, shrinking rate $\gamma \in (0, 1)$, number of trials $l$.
1: **for** $t = 1$ to $T$ **do**
2:     Compute the gradient estimator $g_t$.
3:     $x_t = \arg\min_{x \in \mathcal{X}_t} f(x)$, where $\mathcal{X}_t = \{x_{t-1}, x_{t-1} - \eta \gamma g_t, ..., x_{t-1} - \eta \gamma^{l-1} g_t\}$.
4: **end for**

---

## 4 EXPERIMENTS

### 4.1 SIMPLE FUNCTIONS

In this section, we present experimental results demonstrating the effectiveness of Algorithm 1 on two simple functions: *(1)* Quadratic function: $f(x) = \|x\|_2^2$, $x \in \mathbb{R}^{100}$. *(2)* Rosenbrock function: $f(x) = \sum_{i=1}^{99} \left((1 - x_i)^2 + 100(x_{i+1} - x_i^2)^2\right)$, $x =$

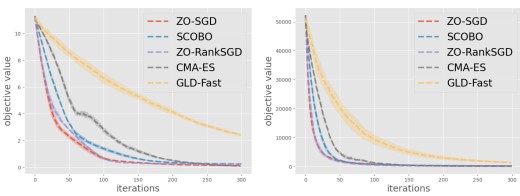

(a) Quadratic function    (b) Rosenbrock function

Figure 3: Performance of different algorithms.

$[x_1, ..., x_{100}]^\top \in \mathbb{R}^{100}$. To demonstrate the effectiveness of our algorithm and verify our theoretical claims, we conduct two experiments, and all figures are obtained by averaging over 10 independent runs and are visualized in the form of mean±std.

**Comparing Algorithm 1 with existing algortihms.** In this first experiment, we compare Algorithm 1 with the following algorithms in the existing literature: *(1)* ZO-SGD (Nesterov & Spokoiny, 2017): A zeroth-order optimization algorithm for valuing oracle. *(2)* SCOBO (Cai et al., 2022): A zeroth-order algorithm for pairwise comparing oracle. *(3)* GLD-Fast (Golovin et al., 2019): A direct search algorithm for top-1 oracle, namely, $(m, 1)$-ranking oracle. *(4)* CMA-ES (Loshchilov & Hutter, 2016; Hansen et al., 2019): A heuristic optimization algorithm for ranking oracle.

To ensure a meaningful comparison, we fix the number of queries $m = 15$ at each iteration for all algorithms. For gradient-based algorithms, ZO-SGD, SCOBO, and our ZO-RankSGD, we use

query points for gradient estimation and 5 points for the line search. In this experiment, we set $m = k$ for ZO-RankSGD, i.e. it can receive the full ranking information. Moreover, we tune the hyperparameters such as stepsize, smoothing parameter, and line search parameter via grid search for each algorithm, and the details are provided in Appendix C.1. A high-dimensional experiment is also included in Appendix C.2. Our experiment results in Figure 3 on the two functions show that the gradient-based algorithm can outperform the direct search algorithm GLD-Fast and the heuristic algorithm CMA-ES. Besides, Algorithm 1 can outperform SCOBO because the ranking oracle contains more information than the pairwise comparison oracle. Additionally, Algorithm 1 behaves similarly to ZO-SGD, indicating that the ranking oracle can be almost as informative as the valuing oracle for zeroth-order optimization.

**Investigating the impact of $m$ and $k$ on Algorithm 1.** In this part, we aim to validate the findings presented in Lemma 4 and Theorem 1 by running Algorithm 1 with various values of $m$ and $k$. To keep the setup simple, we set the step size $\eta$ to 50 and the smoothing parameter $\mu$ to 0.01 for Algorithm 1 with line search (where $l = 5$ and $\gamma = 0.1$). Figure 4 illustrates the performance of ZO-RankSGD under different combinations of $m$ and $k$ on the two functions, which confirm our theoretical findings presented in Lemma 4. For example, we observe that $(m = 10, k = 10)$ yields better performance than $(m = 100, k = 1)$, as predicted by the second variance term in (10), which dominates and scales as $\mathcal{O}(1/k)$.

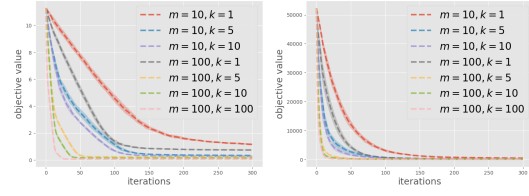

(a) Quadratic function     (b) Rosenbrock function

Figure 4: Performance of ZO-RankSGD under different combinations of $m$ and $k$.

**Noisy ranking oracles.** In practice, the ranking feedback we receive from human evaluators may have some mistakes or inaccuracies. In Appendix C.3, we also present experimental results with noisy ranking oracles, namely, these oracles do not always give the correct ranking feedback. Remarkably, our proposed ZO-RankSGD algorithm shows robustness in handling this kind of noisy feedback, making it well-suited for real-world applications. Looking forward, an interesting area for further exploration is the theoretical understanding of these noisy ranking oracles. This involves understanding how to formally represent the inaccuracy inherent in such oracles, similar to what (Cai et al., 2022) did for comparison oracles.

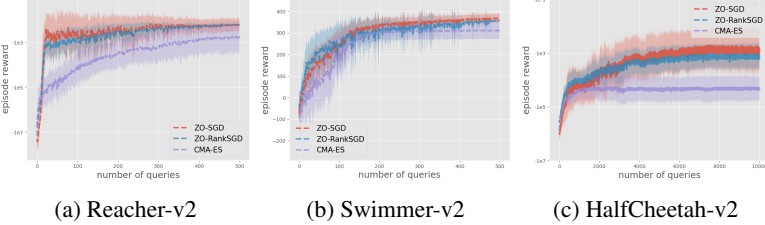

(a) Reacher-v2     (b) Swimmer-v2     (c) HalfCheetah-v2

Figure 5: Perfomance of ZO-RankSGD and CMA-ES on three MuJoCo environments

## 4.2 REINFORCEMENT LEARNING WITH RANKING ORACLES

**Motivation.** In this section, we illustrate how ZO-RankSGD can be seamlessly employed for policy optimization in reinforcement learning, given only a ranking oracle of the episode reward. Such a setting especially captures the scenario where human evaluators are asked to rank multiple episodes based on their expertise. Specifically, we adopt a similar experimental setup as (Cai et al., 2022; Duan et al., 2016), where the goal is to learn a policy for simulated robot control with several problems from the MuJoCo suite of benchmarks (Todorov et al., 2012). We compare ZO-RankSGD to the CMA-ES algorithm, a commonly used optimization baseline in reinforcement learning (Bengs et al., 2021) that also solely relies on a ranking oracle. Both algorithms are restricted to query the episode reward via a $(5, 5)$-ranking oracle. To demonstrate the performance gap between ranking oracle and value oracle, we also include ZO-SGD for comparison. To make a fair comparison, ZO-SGD is designed to receive value feedback of 5 points for each query. Additionally, we draw a comparison between ZO-RankSGD and SCOBO; however, given the disparate nature of their query oracles, the comparison is intricate. For a comprehensive discussion of this aspect and more experiment details, we refer the readers to Appendix C.4.

**Results.** The experiment results are shown in Figure 5, where the x-axis is the number of queries to the ranking oracle, and the y-axis is the ground-truth episode reward. In these experiments, we do not use line search for ZO-RankSGD, instead, we let $\eta = \mu$, and decay them exponentially after every rollout. As can be seen from Figure 5, our algorithm can outperform CMA-ES by a significant margin on all three tasks, exhibiting a better ability to incorporate ranking information. Additionally, ZO-RankSGD exhibits performance on par with ZO-SGD, reinforcing our findings from the experiment illustrated in Figure 3 and underscoring the effectiveness of the ranking oracle in providing substantial optimization-relevant information.

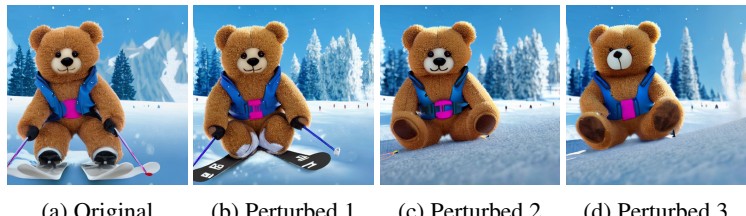

(a) Original  (b) Perturbed 1  (c) Perturbed 2  (d) Perturbed 3

Figure 6: Continuous property of reverse diffusion process. The used text prompt is *A teddy bear is skiing, detailed, realistic, 4K, 3D.*

### 4.3 TAMING DIFFUSION GENERATIVE MODEL WITH HUMAN FEEDBACK

In recent years, there has been a growing interest in diffusion generative models, which have demonstrated remarkable performance in generating high-quality images (Ho et al., 2020; Song et al., 2020b; Dhariwal & Nichol, 2021). Despite these advancements, these models often struggle with capturing intricate details, such as human fingers or key elements in prompts, and sometimes fail to align with user aesthetics. To address this issue, we draw inspiration from recent successes in aligning Language Models with human feedback (Ouyang et al., 2022; Liu et al., 2023; OpenAI, 2022; Bai et al., 2022), and propose to utilize human ranking feedback to enhance the generated images. We noticed a concurrent work (Lee et al., 2023) sharing a similar motivation with us. Specifically, their method is based on the existing approach of RLHF and utilizes a considerable amount of pre-collected data for fine-tuning the diffusion model. Despite this shared motivation, our method tackles a distinct problem from theirs, as our proposed method is not designed for fine-tuning, but to help the model better adapt to the need of new users at inference time, and the human feedback is collected in an online fashion. In this experiment, we aim to demonstrate the ability of ZO-RankSGD to improve the model's output at inference time by optimizing the control variables of generation, such as the latent embedding, with the underlying model fixed. A detailed description is provided below.

**Experimental Setting.** We focus on the task of text-to-image generation, using the state-of-the-art Stable Diffusion model (Rombach et al., 2022) to generate images based on given text prompts. Firstly, we observe that a common practice for generating high-quality images in the community of Stable Diffusion is to run the sampling process multiple times with different random seeds, and then pick the optimal one. Inspired by this, we choose to optimize the latent noise embedding, which is equivalent to random seed, using human ranking feedback through our proposed Algorithm 1, with an aim to produce images that are more appealing to humans. This experimental setting offers several advantages, including: *(1)* The latent embedding is a low-dimensional vector and thus requires only a few rounds of human feedback for optimization. *(2)* It can also serve as a data-collecting step before fine-tuning the model. It is also worth noting that any continuous parameter in the diffusion model can be optimized similarly using human feedback.

**Reverse diffusion process as a continuous mapping.** Firstly, we remark that only ODE-based diffusion samplers, like DDIM (Song et al., 2020a) and DPM-solver (Lu et al., 2022), are used in this study, as now the reverse diffusion process will be deterministic and only depends on the latent embedding. We demonstrate that optimizing the latent embedding is a valid continuous optimization problem by showing that, with slight perturbations of the latent embedding, diffusion samplers can usually generate multiple similar images. An example of this phenomenon is in Figure 6, where the first image is generated using a given latent embedding, while the next three images are generated by perturbing this embedding with noise drawn from $\mathcal{N}(0, 0.1I_d)$.

**Examples.** We illustrate several optimization results in Figure 7, where we ourselves provided the human ranking feedback during these experiments. These instances highlight the improvements

in realism and detail that our proposed Algorithm 1 can bring about through the use of human ranking feedback. To illustrate, in the first example, the image optimized with human guidance portrays human fingers and eyes with enhanced accuracy. In the second example, the optimized image adheres more closely to the prompt instruction, successfully capturing the intended item – orange juice. In the third example, the optimized image delivers a more visually appealing depiction of muscularity. Taken together, these results demonstrate the potential of our approach in refining the quality of generated images using human feedback. For more examples like the ones in Figure 7, and the details of the entire optimization process, we refer the readers to Appendix C.5.

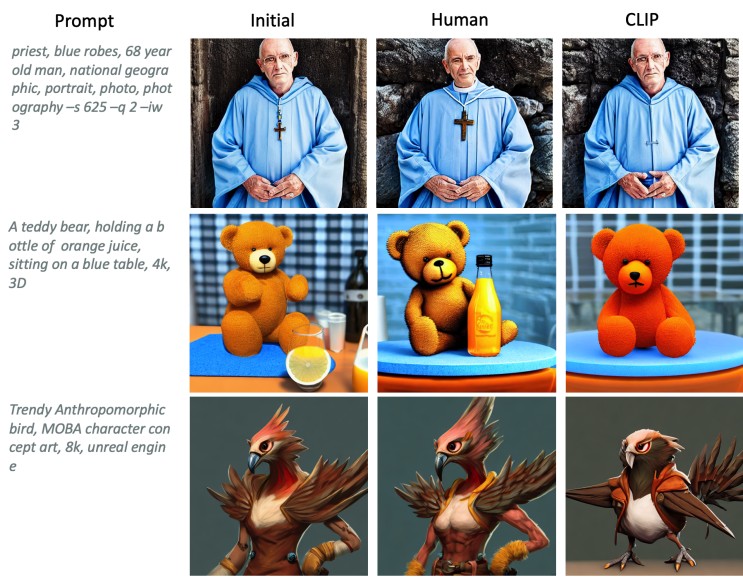

Figure 7: Examples of optimizing latent embedding in diffusion generative model. Initial: The initial images selected through multiple randomly generated latent embeddings serve as the initial points for the later optimization process. Human: The images obtained by optimizing human preference. CLIP: The images obtained by optimizing the CLIP similarity score.

**Human feedback vs. CLIP similarity score.** To underscore the unique advantage of human feedback, we hold the ZO-RankSGD algorithm constant, and contrast images that were optimized with human preference against those optimized using the CLIP similarity score (Radford et al., 2021). CLIP, a cutting-edge model that contrasts language with images, calculates the similarity between given texts and images. However, when comparing the third and fourth columns in Figure 7, it is clear that since CLIP is trained on noisy text-image pairs from the internet, the images optimized using its similarity score can sometimes fall short of the original ones. Moreover, these CLIP-optimized images may not always resonate with human evaluators, further emphasizing the unique value of human feedback in refining image generation.

## 5 CONCLUSION

In this paper, we have rigorously studied a novel optimization problem where only ranking oracles of the objective function are available. For this problem, we have proposed the first provable zeroth-order optimization algorithm, ZO-RankSGD, which has consistently demonstrated its efficacy across simulated and real-world applications. We also have presented how different ranking oracles can impact optimization performance, providing guidance on designing the user interface for ranking feedback. Our algorithm has been shown to be a practical and effective way to incorporate human feedback, for example, it can be used to improve the detail of images generated by Stable Diffusion with human guidance. Possible future directions to this work may include extending the theoretical results to incorporate noisy and uncertain ranking feedback, combining ZO-RankSGD with a model-based approach like Bayesian Optimization (Frazier, 2018), or the techniques from active learning (Monarch, 2021), to further improve the query efficiency, and also applying it to other scenarios beyond human feedback. Besides, another important point is to investigate what is the optimal choice of $m$ and $k$ if we jointly consider the cognitive burden of humans and the query complexity via real social experiments.

## ACKNOWLEDGMENTS

The work is supported by Shenzhen Science and Technology Program under Grant No. RCJC20210609104448114, the NSFC, China, under Grant 62071409, and by Guangdong Provincial Key Laboratory of Big Data Computing.

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

## APPENDIX

### A    A SIMPLIFIED EXPRESSION FOR (5)

Let $\mathcal{G} = (\mathcal{N}, \mathcal{E})$ be the DAG constructed from the ranking information of $O_f^{(m,k)}$, we denote the input degrees and output degrees of $x_i \in \mathcal{N}$ as $\deg_{\text{in}}(i)$ and $\deg_{\text{out}}(i)$ respectively. We first notice that

$$\sum_{(i,j) \in \mathcal{E}} (\xi_j - \xi_i) = \sum_{i=1}^{m} (\deg_{\text{in}}(i) - \deg_{\text{out}}(i)) \, \xi_i. \tag{14}$$

Denote $w_i = \deg_{\text{in}}(i) - \deg_{\text{out}}(i)$, if $O_f^{(m,k)}(x_1, ..., x_m) = (i_1, ..., i_k)$, then we can compute that

$$w_{i_j} = \deg_{\text{in}}(i_j) - \deg_{\text{out}}(i_j) = j - 1 - (m - j) = 2j - m - 1, \quad j = 1, ..., k. \tag{15}$$

$$w_q = \deg_{\text{in}}(q) - \deg_{\text{out}}(q) = k - 0 = k, \quad q \notin \{i_1, ..., i_k\}. \tag{16}$$

### B    MISSING PROOF

*Proof of Lemma 1.* In the following proof, we denote $p(\cdot)$ as the pdf function of $\mathcal{N}(0, I_d)$ for arbitrary dimension $d$.

We first rewrite $\langle \nabla f(x), \hat{g}(x) \rangle$ as follows:

$$\langle \nabla f(x), \hat{g}(x) \rangle = \langle \nabla f(x), S_f(x, \xi_1, \xi_2, \mu)(\xi_1 - \xi_2) \rangle = S_f(x, \xi_1, \xi_2, \mu) \cdot \langle \nabla f(x), \xi_1 - \xi_2 \rangle. \tag{17}$$

By the second-order Taylor expansion with Cauchy remainders, we notice that

$$f(x + \mu \xi_1) = f(x) + \mu \langle \nabla f(x), \xi_1 \rangle + \frac{\mu^2}{2} \xi_1^\top \nabla^2 f(x_1) \xi_1, \tag{18}$$

$$f(x + \mu \xi_2) = f(x) + \mu \langle \nabla f(x), \xi_2 \rangle + \frac{\mu^2}{2} \xi_2^\top \nabla^2 f(x_2) \xi_2, \tag{19}$$

where $x_1$ and $x_2$ are two points around $x$.

With (18) and (19) we can write $S_f(x, \xi_1, \xi_2, \mu)$ as follows:

$$S_f(x, \xi_1, \xi_2, \mu) = \text{Sign}\left(\langle \nabla f(x), \xi_1 - \xi_2 \rangle + \frac{\mu}{2}\xi_1^\top \nabla^2 f(x_1)\xi_1 - \frac{\mu}{2}\xi_2^\top \nabla^2 f(x_2)\xi_2\right). \quad (20)$$

Now we start to bound the term

$$\mathbb{E}\left[S_f(x, \xi_1, \xi_2, \mu) \cdot \langle \nabla f(x), \xi_1 - \xi_2 \rangle\right], \quad (21)$$

where the expectation is taken over the random direction $\xi_1$ and $\xi_2$.

Before doing that, we first define two important regions:

$$\mathcal{R}_1 = \{(\xi_1, \xi_2) \mid \langle \nabla f(x), \xi_1 - \xi_2 \rangle > 0\}, \quad (22)$$

$$\mathcal{R}_{11} = \{(\xi_1, \xi_2) \mid (\xi_1, \xi_2) \in \mathcal{R}_1, S_f(x, \xi_1, \xi_2, \mu) \neq \text{Sign}(\langle \nabla f(x), \xi_1 - \xi_2 \rangle)\}. \quad (23)$$

Notice that when $(\xi_1, \xi_2) \in \mathcal{R}_1$, $S_f(x, \xi_1, \xi_2, \mu) \neq \text{Sign}(\langle \nabla f(x), \xi_1 - \xi_2 \rangle)$ is equivalent to

$$\langle \nabla f(x), \xi_1 - \xi_2 \rangle + \frac{\mu}{2}\xi_1^\top \nabla^2 f(x_1)\xi_1 - \frac{\mu}{2}\xi_2^\top \nabla^2 f(x_2)\xi_2 < 0.$$

Also, from $L$-smoothness, we can know that

$$-\frac{\mu L}{2}\left(\|\xi_1\|_2^2 + \|\xi_2\|_2^2\right) \leq \frac{\mu}{2}\xi_1^\top \nabla^2 f(x_1)\xi_1 - \frac{\mu}{2}\xi_2^\top \nabla^2 f(x_2)\xi_2.$$

We denote the region

$$\bar{\mathcal{R}}_{11} = \{(\xi_1, \xi_2) \mid (\xi_1, \xi_2) \in \mathcal{R}_1, \langle \nabla f(x), \xi_1 - \xi_2 \rangle - \frac{\mu L}{2}\left(\|\xi_1\|_2^2 + \|\xi_2\|_2^2\right) < 0\}. \quad (24)$$

It is easy to verify that $\mathcal{R}_{11} \subseteq \bar{\mathcal{R}}_{11}$. Let $\mathcal{R}_{12} = \mathcal{R}_1/\bar{\mathcal{R}}_{11}$, we can have the following inequality.

$$\int_{\mathcal{R}_1} S_f(x, \xi_1, \xi_2, \mu) \langle \nabla f(x), \xi_1 - \xi_2 \rangle p(\xi_1)p(\xi_2)d\xi_1 d\xi_2 \quad (25)$$

$$= \int_{\mathcal{R}_1/\mathcal{R}_{11}} S_f(x, \xi_1, \xi_2, \mu) \langle \nabla f(x), \xi_1 - \xi_2 \rangle p(\xi_1)p(\xi_2)d\xi_1 d\xi_2$$

$$+ \int_{\mathcal{R}_{11}} S_f(x, \xi_1, \xi_2, \mu) \langle \nabla f(x), \xi_1 - \xi_2 \rangle p(\xi_1)p(\xi_2)d\xi_1 d\xi_2 \quad (26)$$

$$= \int_{\mathcal{R}_1/\mathcal{R}_{11}} \langle \nabla f(x), \xi_1 - \xi_2 \rangle p(\xi_1)p(\xi_2)d\xi_1 d\xi_2$$

$$- \int_{\mathcal{R}_{11}} \langle \nabla f(x), \xi_1 - \xi_2 \rangle p(\xi_1)p(\xi_2)d\xi_1 d\xi_2 \quad (27)$$

$$\geq \int_{\mathcal{R}_1/\bar{\mathcal{R}}_{11}} \langle \nabla f(x), \xi_1 - \xi_2 \rangle p(\xi_1)p(\xi_2)d\xi_1 d\xi_2$$

$$- \int_{\bar{\mathcal{R}}_{11}} \langle \nabla f(x), \xi_1 - \xi_2 \rangle p(\xi_1)p(\xi_2)d\xi_1 d\xi_2 \quad (28)$$

$$= 2\int_{\mathcal{R}_{12}} \langle \nabla f(x), \xi_1 - \xi_2 \rangle p(\xi_1)p(\xi_2)d\xi_1 d\xi_2$$

$$- \int_{\mathcal{R}_1} \langle \nabla f(x), \xi_1 - \xi_2 \rangle p(\xi_1)p(\xi_2)d\xi_1 d\xi_2. \quad (29)$$

Before we proceed to study the property of the integral in (29), let us first define an important function. Consider the function $h(v, r, d) : \mathbb{R} \times \mathbb{R}_+ \times \mathbb{Z}_+ \to \mathbb{R}$ defined as follows:

$$h(v, r, d) \stackrel{\text{def.}}{=} \sqrt{2}v \int_0^{\frac{2\sqrt{2}v}{r}} xF_{2d-1}\left(\left(\frac{2\sqrt{2}v}{r} - x\right)x\right) p(x)dx, \quad (30)$$

where $F_{2d-1}(\cdot)$ is the CDF of the $\chi^2$ distribution with $2d-1$ degrees of freedom. With this function, we can have the following lemma that presents the close form of the integrals in (29).

**Lemma 5.**

$$\int_{\mathcal{R}_1} \langle \nabla f(x), \xi_1 - \xi_2 \rangle \, p(\xi_1) p(\xi_2) d\xi_1 d\xi_2 = \frac{1}{\sqrt{\pi}} \|\nabla f(x)\|, \tag{31}$$

$$\int_{\mathcal{R}_{12}} \langle \nabla f(x), \xi_1 - \xi_2 \rangle \, p(\xi_1) p(\xi_2) d\xi_1 d\xi_2 = h(\|\nabla f(x)\|, \mu L, d). \tag{32}$$

Also, we need an important lemma on $h(v, r, d)$.

**Lemma 6.** *For any $d \in \mathbb{Z}_+$, there exist a constant $C_d > 0$ such that for any $v \geq 0$, $r > 0$,*

$$h(v, r, d) \geq \left( \frac{1}{2\sqrt{\pi}} + \frac{1}{4} \right) v - \frac{1}{4} C_d r. \tag{33}$$

Combining (29), (31), (32) and (33), we have

$$\int_{\mathcal{R}_1} S_f(x, \xi_1, \xi_2, \mu) \langle \nabla f(x), \xi_1 - \xi_2 \rangle \, p(\xi_1) p(\xi_2) d\xi_1 d\xi_2 \geq \frac{1}{2} \|\nabla f(x)\| - \frac{1}{2} C_d \mu L. \tag{34}$$

Similarly, if we define

$$\mathcal{R}_2 = \{ (\xi_1, \xi_2) \mid \langle \nabla f(x), \xi_1 - \xi_2 \rangle < 0 \},$$

we have

$$\int_{\mathcal{R}_2} S_f(x, \xi_1, \xi_2, \mu) \langle \nabla f(x), \xi_1 - \xi_2 \rangle \, p(\xi_1) p(\xi_2) d\xi_1 d\xi_2 \tag{35}$$

$$= \int_{\mathcal{R}_2} S_f(x, \xi_2, \xi_1, \mu) \langle \nabla f(x), \xi_2 - \xi_1 \rangle \, p(\xi_1) p(\xi_2) d\xi_1 d\xi_2 \tag{36}$$

$$= \int_{\mathcal{R}_1} S_f(x, \xi_1, \xi_2, \mu) \langle \nabla f(x), \xi_1 - \xi_2 \rangle \, p(\xi_1) p(\xi_2) d\xi_1 d\xi_2, \tag{37}$$

becasue the integral on $\mathcal{R}_1$ is symmetric to the integral on $\mathcal{R}_2$ by swapping $\xi_1$ and $\xi_2$. Since $\mathbb{R}^{2d} / (\mathcal{R}_1 \cup \mathcal{R}_2)$ has zero measure, we have

$$\mathbb{E} \left[ S_f(x, \xi_1, \xi_2, \mu) \cdot \langle \nabla f(x), \xi_1 - \xi_2 \rangle \right]$$

$$= 2 \int_{\mathcal{R}_1} \langle \nabla f(x), \xi_2 - \xi_1 \rangle \, p(\xi_1) p(\xi_2) d\xi_1 d\xi_2 \tag{38}$$

$$\geq \|\nabla f(x)\| - C_d \mu L. \tag{39}$$

$\square$

*Proof of Lemma 2.* Suppose that $O_f^{(m,k)}(x_1, ..., x_m) = (i_1, ..., i_k)$, we seperate $\mathcal{N}$ into two node set:

$$\mathcal{N}_1 = \{ i_1, ..., i_k \} \text{ and } \mathcal{N}_2 = \{ q \in \{1, ..., m\} \mid q \notin \{i_1, ..., i_k\} \}.$$

Firstly, since the subgraph of $\mathcal{G}$ on $\mathcal{N}_1$ is a complete graph, the number of edges in this subgraph is $k(k-1)/2$. The remaining edges in $\mathcal{G}$ connect the node in $\mathcal{N}_2$ to the node in $\mathcal{N}_1$, hence the number of them is $k(m-k)$. Therefore,

$$|\mathcal{E}| = k(k-1)/2 + k(m-k) = km - (k^2 + k)/2. \tag{40}$$

Now we denote the set of neighbooring edge pairs as $\mathcal{S} = \{ ((i, j), (i', j)) \in \bar{\mathcal{E}} \times \bar{\mathcal{E}} \mid i \neq i' \}$. We can split $\mathcal{S}$ as the following five set:

$$S_1 = \{ ((i, j), (i', j)) \in \bar{\mathcal{E}} \times \bar{\mathcal{E}} \mid i \neq i', i \in \mathcal{N}_1, i' \in \mathcal{N}_1, j \in \mathcal{N}_1 \}, \tag{41}$$

$$S_2 = \{ ((i, j), (i', j)) \in \bar{\mathcal{E}} \times \bar{\mathcal{E}} \mid i \neq i', i \in \mathcal{N}_1, i' \in \mathcal{N}_1, j \in \mathcal{N}_2 \}, \tag{42}$$

$$S_3 = \{ ((i, j), (i', j)) \in \bar{\mathcal{E}} \times \bar{\mathcal{E}} \mid i \neq i', i \in \mathcal{N}_1, i' \in \mathcal{N}_2, j \in \mathcal{N}_1 \}, \tag{43}$$

$$S_4 = \{ ((i, j), (i', j)) \in \bar{\mathcal{E}} \times \bar{\mathcal{E}} \mid i \neq i', i \in \mathcal{N}_2, i' \in \mathcal{N}_1, j \in \mathcal{N}_1 \}, \tag{44}$$

$$S_5 = \{ ((i, j), (i', j)) \in \bar{\mathcal{E}} \times \bar{\mathcal{E}} \mid i \neq i', i \in \mathcal{N}_2, i' \in \mathcal{N}_2, j \in \mathcal{N}_1 \}. \tag{45}$$

For the first set $\mathcal{S}_1$, we can compute that

$$|\mathcal{S}_1| = 6\binom{k}{3} = k(k-1)(k-2), \tag{46}$$

because every edge pair composes of three nodes, and every three nodes can form 6 edge pairs.

For the second set $\mathcal{S}_2$, we have

$$|\mathcal{S}_2| = 2(m-k)\binom{k}{2} = (m-k)k(k-1), \tag{47}$$

because $|\mathcal{N}_2| = m - k$ and $|\{(i, i') \in \mathcal{N}_1 \times \mathcal{N}_1 \mid i \neq i'\}| = 2\binom{k}{2}$.

Similarly, for the set $\mathcal{S}_3$ and $\mathcal{S}_4$, we can obtain

$$|\mathcal{S}_3| = |\mathcal{S}_4| = 2(m-k)\binom{k}{2} = (m-k)k(k-1). \tag{48}$$

Finally, for the set $\mathcal{S}_5$, we can compute that

$$|\mathcal{S}_5| = 2k\binom{m-k}{2} = k(m-k)(m-k-1), \tag{49}$$

because $|\mathcal{N}_1| = k$ and $|\{(i, i') \in \mathcal{N}_2 \times \mathcal{N}_2 \mid i \neq i'\}| = 2\binom{m-k}{2}$.

In all, we have

$$|\mathcal{S}| = |\mathcal{S}_1| + |\mathcal{S}_2| + |\mathcal{S}_3| + |\mathcal{S}_4| + |\mathcal{S}_5| \tag{50}$$
$$= k(k-1)(k-2) + 3(m-k)k(k-1) + k(m-k)(m-k-1) \tag{51}$$
$$= m^2 k + mk^2 - k^3 + k^2 - 4mk + 2k. \tag{52}$$

$\square$

*Proof of Lemma 3.* We first prove that $M_1(f, \mu) \leq 2d$. From convexity of $\|\cdot\|^2$ and Jensen's inequality, we have

$$\left\| \mathbb{E}_{\xi_1, \xi_2} [S_f(x, \xi_1, \xi_2, \mu)(\xi_1 - \xi_2)] \right\|^2 \leq \mathbb{E}_{\xi_1, \xi_2} \|[S_f(x, \xi_1, \xi_2, \mu)(\xi_1 - \xi_2)]\|^2 = 2d. \tag{53}$$

Then we prove $M_2(f, \mu) \leq 2d$. From the Cauchy-Schwarz inequality, we have

$$\mathbb{E}_{\xi_1, \xi_2, \xi_3} [S_f(x, \xi_1, \xi_2, \mu) S_f(x, \xi_1, \xi_3, \mu) \langle \xi_1 - \xi_2, \xi_1 - \xi_3 \rangle] \tag{54}$$

$$\leq \sqrt{\mathbb{E}_{\xi_1, \xi_2} \left[ \|\xi_1 - \xi_2\|^2 \right] \mathbb{E}_{\xi_1, \xi_3} \left[ \|\xi_1 - \xi_3\|^2 \right]} = 2d. \tag{55}$$

Now we study the mean vector $\mathbb{E}_{\xi_1, \xi_2} [S_f(x, \xi_1, \xi_2, \mu)(\xi_1 - \xi_2)]$ under the condition $\nabla^2 f(x) = cI_d$. We first write it as a sum of three vectors.

$$\mathbb{E}_{\xi_1, \xi_2} [S_f(x, \xi_1, \xi_2, \mu)(\xi_1 - \xi_2)] = \int_{f(x+\mu\xi_1) > f(x+\mu\xi_2)} (\xi_1 - \xi_2) p(\xi_1) p(\xi_2) d\xi_1 d\xi_2 \tag{56}$$

$$+ \int_{f(x+\mu\xi_1) = f(x+\mu\xi_2)} (\xi_1 - \xi_2) p(\xi_1) p(\xi_2) d\xi_1 d\xi_2 \tag{57}$$

$$+ \int_{f(x+\mu\xi_1) < f(x+\mu\xi_2)} (\xi_2 - \xi_1) p(\xi_1) p(\xi_2) d\xi_1 d\xi_2. \tag{58}$$

For the three vectors, we have

$$\int_{f(x+\mu\xi_1) = f(x+\mu\xi_2)} (\xi_1 - \xi_2) p(\xi_1) p(\xi_2) d\xi_1 d\xi_2 \tag{59}$$

$$= \int_{f(x+\mu\xi_1) = f(x+\mu\xi_2)} \xi_1 p(\xi_1) p(\xi_2) d\xi_1 d\xi_2 - \int_{f(x+\mu\xi_1) = f(x+\mu\xi_2)} \xi_2 p(\xi_1) p(\xi_2) d\xi_1 d\xi_2 \tag{60}$$

$$= 0, \tag{61}$$

and

$$\int_{f(x+\mu\xi_1)>f(x+\mu\xi_2)} (\xi_1 - \xi_2)p(\xi_1)p(\xi_2)d\xi_1 d\xi_2 \tag{62}$$

$$= \int_{f(x+\mu\xi_2)>f(x+\mu\xi_1)} (\xi_2 - \xi_1)p(\xi_1)p(\xi_2)d\xi_1 d\xi_2 \tag{63}$$

$$= \int_{f(x+\mu\xi_1)<f(x+\mu\xi_2)} (\xi_2 - \xi_1)p(\xi_1)p(\xi_2)d\xi_1 d\xi_2. \tag{64}$$

Therefore, we can write $\mathbb{E}_{\xi_1,\xi_2}\left[S_f(x,\xi_1,\xi_2,\mu)(\xi_1-\xi_2)\right]$ as

$$\mathbb{E}_{\xi_1,\xi_2}\left[S_f(x,\xi_1,\xi_2,\mu)(\xi_1-\xi_2)\right] = 2\int_{f(x+\mu\xi_1)>f(x+\mu\xi_2)} (\xi_1 - \xi_2)p(\xi_1)p(\xi_2)d\xi_1 d\xi_2. \tag{65}$$

Now we study the integrals $\int_{f(x+\mu\xi_1)>f(x+\mu\xi_2)} \xi_1 p(\xi_1)p(\xi_2)d\xi_1 d\xi_2$ and $\int_{f(x+\mu\xi_1)>f(x+\mu\xi_2)} \xi_2 p(\xi_1)p(\xi_2)d\xi_1 d\xi_2$. We can compute that

$$\int_{f(x+\mu\xi_1)>f(x+\mu\xi_2)} \xi_1 p(\xi_1)p(\xi_2)d\xi_1 d\xi_2 \tag{66}$$

$$= \int_{\mathbb{R}^d} \left( \int_{f(x+\mu\xi_1)>f(x+\mu\xi_2)} p(\xi_2)d\xi_2 \right) \xi_1 p(\xi_1)d\xi_1, \tag{67}$$

and,

$$\int_{f(x+\mu\xi_1)>f(x+\mu\xi_2)} \xi_2 p(\xi_1)p(\xi_2)d\xi_1 d\xi_2 \tag{68}$$

$$= \int_{\mathbb{R}^d} \left( \int_{f(x+\mu\xi_1)>f(x+\mu\xi_2)} p(\xi_1)d\xi_1 \right) \xi_2 p(\xi_2)d\xi_2 \tag{69}$$

$$= \int_{\mathbb{R}^d} \left( \int_{f(x+\mu\xi_2)>f(x+\mu\xi_1)} p(\xi_2)d\xi_2 \right) \xi_1 p(\xi_1)d\xi_1 \tag{70}$$

The condition $\nabla^2 f(x) = cI_d$ implies that $f$ is a quadratic function. We denote $\mathcal{M}(\cdot)$ as the Lebesgue measure on $\mathbb{R}^d$. Notice that $\mathcal{M}(\{\xi_2 \mid f(x+\mu\xi_2) = f(x+\mu\xi_1)\}) = 0$ because it is known that the zero point set of any polynomial function has zero Lebesgue measure. Therefore, we have

$$\int_{f(x+\mu\xi_1)>f(x+\mu\xi_2)} p(\xi_2)d\xi_2 + \int_{f(x+\mu\xi_2)>f(x+\mu\xi_1)} p(\xi_2)d\xi_2 \tag{71}$$

$$= 1 - \int_{f(x+\mu\xi_2)=f(x+\mu\xi_1)} p(\xi_2)d\xi_2 = 1. \tag{72}$$

Hence we have

$$\int_{f(x+\mu\xi_1)>f(x+\mu\xi_2)} (\xi_1 - \xi_2)p(\xi_1)p(\xi_2)d\xi_1 d\xi_2 \tag{73}$$

$$= 2\int_{\mathbb{R}^d} \left( \int_{f(x+\mu\xi_1)>f(x+\mu\xi_2)} p(\xi_2)d\xi_2 \right) \xi_1 p(\xi_1)d\xi_1 - \int_{\mathbb{R}^d} \xi_1 p(\xi_1)d\xi_1 \tag{74}$$

$$= 2\int_{\mathbb{R}^d} \left( \int_{f(x+\mu\xi_1)>f(x+\mu\xi_2)} p(\xi_2)d\xi_2 \right) \xi_1 p(\xi_1)d\xi_1. \tag{75}$$

Since $\nabla^2 f(x) = cI_d$, we have

$$f(x+\mu\xi_1) = f(x) + \mu\nabla f(x)^T \xi_1 + \frac{1}{2}\mu^2 \|\xi_1\|^2.$$

Without loss of generality, we assume $\|\nabla f(x)\| \neq 0$ and denote $\xi'_1 = \frac{2\langle \nabla f(x), \xi_1 \rangle}{\|\nabla f(x)\|^2} \nabla f(x) - \xi_1$. It is easy to verify that $\xi'_1$ also follows $\mathcal{N}(0, I_d)$, $\|\xi'_1\| = \|\xi_1\|$ and $f(x + \mu \xi_1) = f(x + \mu \xi'_1)$. Therefore, we have

$$\int_{\mathbb{R}^d} \left( \int_{f(x+\mu\xi_1) > f(x+\mu\xi_2)} p(\xi_2) d\xi_2 \right) \xi_1 p(\xi_1) d\xi_1 \tag{76}$$

$$= \int_{\mathbb{R}^d} \left( \int_{f(x+\mu\xi_1) > f(x+\mu\xi_2)} p(\xi_2) d\xi_2 \right) \xi'_1 p(\xi'_1) d\xi'_1 \tag{77}$$

$$= \frac{1}{2} \int_{\mathbb{R}^d} \left( \int_{f(x+\mu\xi_1) > f(x+\mu\xi_2)} p(\xi_2) d\xi_2 \right) (\xi_1 + \xi') \, p(\xi_1) d\xi_1. \tag{78}$$

$$= \left( \int_{\mathbb{R}^d} \left( \int_{f(x+\mu\xi_1) > f(x+\mu\xi_2)} p(\xi_2) d\xi_2 \right) \frac{\langle \nabla f(x), \xi_1 \rangle}{\|\nabla f(x)\|} p(\xi_1) d\xi_1 \right) \frac{\nabla f(x)}{\|\nabla f(x)\|}. \tag{79}$$

Furthermore,

$$\left| \int_{\mathbb{R}^d} \left( \int_{f(x+\mu\xi_1) > f(x+\mu\xi_2)} p(\xi_2) d\xi_2 \right) \frac{\langle \nabla f(x), \xi_1 \rangle}{\|\nabla f(x)\|} p(\xi_1) d\xi_1 \right| \leq \int_{\mathbb{R}^d} \frac{|\langle \nabla f(x), \xi_1 \rangle|}{\|\nabla f(x)\|} p(\xi_1) d\xi_1 = \sqrt{\frac{2}{\pi}}. \tag{80}$$

Finally, we have

$$\left\| \mathbb{E}_{\xi_1, \xi_2} \left[ S_f(x, \xi_1, \xi_2, \mu)(\xi_1 - \xi_2) \right] \right\|^2 \tag{81}$$

$$= \left\| 4 \left( \int_{\mathbb{R}^d} \left( \int_{f(x+\mu\xi_1) > f(x+\mu\xi_2)} p(\xi_2) d\xi_2 \right) \frac{\langle \nabla f(x), \xi_1 \rangle}{\|\nabla f(x)\|} p(\xi_1) d\xi_1 \right) \frac{\nabla f(x)}{\|\nabla f(x)\|} \right\|^2 \tag{82}$$

$$\leq \frac{32}{\pi}. \tag{83}$$

$$\square$$

*Proof of Lemma 4.* We first compute that

$$E \left[ \|\tilde{g}(x)\|_2^2 \right] = \frac{1}{|\mathcal{E}|^2} E \left[ \left\| \sum_{(i,j) \in \mathcal{E}} (\xi_j - \xi_i) \right\|_2^2 \right]. \tag{84}$$

For ease of writing, we denote $B_{(i,j)} = \xi_j - \xi_i = S_f(x, \xi_i, \xi_j, \mu)(\xi_i - \xi_j)$ and $\bar{\mathcal{E}}$ as the undirected version of $\mathcal{E}$.

$$E \left[ \left\| \sum_{(i,j) \in \mathcal{E}} B_{(i,j)} \right\|_2^2 \right] \tag{85}$$

$$= E \left[ \sum_{(i,j) \in \mathcal{E}} \left\| B_{(i,j)} \right\|_2^2 + \sum_{\substack{(i,j) \in \bar{\mathcal{E}} \\ (i',j) \in \bar{\mathcal{E}} \\ i \neq i'}} \left\langle B_{(i,j)}, B_{(i',j)} \right\rangle + \sum_{\substack{(i,j) \in \bar{\mathcal{E}} \\ (i',j') \in \bar{\mathcal{E}} \\ i \neq i', j \neq j'}} \left\langle B_{(i,j)}, B_{(i',j')} \right\rangle \right]. \tag{86}$$

With the two metrics $M_1(f, \mu)$, $M_2(f, \mu)$, we can bound the four terms in (86) as follows:

$$E\left[\left\|B_{(i,j)}\right\|_2^2\right] = E\left[\left\|\xi_j - \xi_i\right\|_2^2\right] = 2d, \tag{87}$$

$$E\left[\langle B_{(i,j)}, B_{(i',j)}\rangle\right] = E\left[\langle B_{(i,j)}, B_{(i,j')}\rangle\right] \leq M_2(f, \mu), \tag{88}$$

$$E\left[\langle B_{(i,j)}, B_{(i',j')}\rangle\right] = \left\|E\left[B_{(i,j)}\right]\right\|_2^2 \leq M_1(f, \mu). \tag{89}$$

Taking (87), (88) and (89) into (86), we obtain

$$E\left[\left\|\sum_{(i,j)\in\mathcal{E}} B_{(i,j)}\right\|_2^2\right] \tag{90}$$

$$\leq \sum_{(i,j)\in\mathcal{E}} 2d + \sum_{\substack{(i,j)\in\bar{\mathcal{E}} \\ (i',j)\in\bar{\mathcal{E}} \\ i\neq i'}} M_2(f, \mu) + \sum_{\substack{(i,j)\in\bar{\mathcal{E}} \\ (i',j')\in\bar{\mathcal{E}} \\ i\neq i', j\neq j'}} M_1(f, \mu) \tag{91}$$

$$= 2|\mathcal{E}|d + N(\mathcal{E})M_2(f, \mu) + (|\mathcal{E}|^2 - N(\mathcal{E}) - |\mathcal{E}|)M_1(f, \mu). \tag{92}$$

Combing (92) with (84), we obtain

$$E\left[\|\tilde{g}(x)\|_2^2\right] \leq \frac{2d}{|\mathcal{E}|} + \frac{N(\mathcal{E})}{|\mathcal{E}|^2}M_2(f, \mu) + \frac{|\mathcal{E}|^2 - N(\mathcal{E}) - |\mathcal{E}|}{|\mathcal{E}|^2}M_1(f, \mu) \tag{93}$$

$$\leq \frac{2d}{|\mathcal{E}|} + \frac{N(\mathcal{E})}{|\mathcal{E}|^2}M_2(f, \mu) + M_1(f, \mu). \tag{94}$$

$\square$

*Proof of Theorem 1.* Consider the $t$-th iteration, from $L$-smoothness we know that

$$f(x_t) - f(x_{t-1}) \leq -\eta\langle\nabla f(x_{t-1}), g_t\rangle + \frac{\eta^2 L}{2}\|g_t\|_2^2. \tag{95}$$

Using Lemma 1 and Lemma 4, we have

$$\mathbb{E}[f(x_t) - f(x_{t-1})] \leq -\eta\langle\nabla f(x_{t-1}), E[g_t]\rangle + \frac{\eta^2 L}{2}E\left[\|g_t\|_2^2\right] \tag{96}$$

$$\leq -\eta\|\nabla f(x_{t-1})\| + C_d\eta\mu L + \frac{\eta^2 L}{2}\left(\frac{2d}{|\mathcal{E}|} + \frac{N(\mathcal{E})}{|\mathcal{E}|^2}M_2(f, \mu) + M_1(f, \mu)\right), \tag{97}$$

where the expectation is taken over the random direction $\xi_{(t,1)}, \cdots, \xi_{(t,m)}$.

Rearrange the inequality to obtain

$$\|\nabla f(x_{t-1})\| \leq \frac{\mathbb{E}[f(x_{t-1}) - f(x_t)]}{\eta} + C_d\mu L + \frac{\eta L}{2}\left(\frac{2d}{|\mathcal{E}|} + \frac{N(\mathcal{E})}{|\mathcal{E}|^2}M_2(f, \mu) + M_1(f, \mu)\right). \tag{98}$$

Summing up over $T$ iterations and dividing both sides by $T$, we finally obtain

$$\mathbb{E}\left[\frac{1}{T}\sum_{t=1}^{T}\|\nabla f(x_{t-1})\|\right] \leq \frac{\mathbb{E}[f(x_0) - f(x_T)]}{\eta T} + C_d\mu L + \frac{\eta L}{2}\left(\frac{2d}{|\mathcal{E}|} + \frac{N(\mathcal{E})}{|\mathcal{E}|^2}M_2(f, \mu) + M_1(f, \mu)\right) \tag{99}$$

$$\leq \frac{f(x_0) - f^*}{\eta T} + C_d\mu L + \frac{\eta L}{2}\left(\frac{2d}{|\mathcal{E}|} + \frac{N(\mathcal{E})}{|\mathcal{E}|^2}M_2(f, \mu) + M_1(f, \mu)\right). \tag{100}$$

The proof is completed by noting that

$$\mathbb{E}\left[\min_{t\in\{1,\dots,T\}}\|\nabla f(x_{t-1})\|\right] \leq \mathbb{E}\left[\frac{1}{T}\sum_{t=1}^{T}\|\nabla f(x_{t-1})\|\right].$$

$\square$

*Proof of Lemma 5.* Without loss of generality, we assume $\|\nabla f(x)\| \neq 0$. We first prove that

$$\int_{\mathcal{R}_1}\langle\nabla f(x),\xi_1-\xi_2\rangle\,p(\xi_1)p(\xi_2)d\xi_1 d\xi_2 = \frac{1}{\sqrt{\pi}}\|\nabla f(x)\|.$$

Now we denote

$$x = \frac{\langle\nabla f(x),\xi_1-\xi_2\rangle}{\sqrt{2}\|\nabla f(x)\|}.$$

Notice that $x$ follows the distribution $\mathcal{N}(0,1)$. Therefore, we have

$$\int_{\mathcal{R}_1}\langle\nabla f(x),\xi_1-\xi_2\rangle\,p(\xi_1)p(\xi_2)d\xi_1 d\xi_2 \tag{101}$$

$$=\sqrt{2}\|\nabla f(x)\|\int_{x>0}xp(x)dx = \frac{1}{\sqrt{\pi}}\|\nabla f(x)\|, \tag{102}$$

where we use a well-known fact that $\int_{x>0}xp(x)dx = \frac{1}{\sqrt{2\pi}}$.

Then we will prove

$$\int_{\mathcal{R}_{12}}\langle\nabla f(x),\xi_1-\xi_2\rangle\,p(\xi_1)p(\xi_2)d\xi_1 d\xi_2 = h(\|\nabla f(x)\|,\mu L,d).$$

Notice that

$$\mathcal{R}_{12} = \{(\xi_1,\xi_2)\mid(\xi_1,\xi_2)\in\mathcal{R}_1, \langle\nabla f(x),\xi_1-\xi_2\rangle - \frac{\mu L}{2}\left(\|\xi_1\|_2^2+\|\xi_2\|_2^2\right)\geq 0\}.$$

We can see that $\mathcal{R}_{12}$ is a ball in $\mathbb{R}^{2d}$:

$$\mathcal{R}_{12} = \left\{(\xi_1,\xi_2)\mid\left\|\xi_1-\frac{1}{\mu L}\nabla f(x)\right\|_2^2+\left\|\xi_2+\frac{1}{\mu L}\nabla f(x)\right\|_2^2 < \frac{2\|\nabla f(x)\|_2^2}{\mu^2 L^2}\right\}. \tag{103}$$

Now we denote $\zeta = [-\xi_1^\top,\xi_2^\top]^\top \in \mathbb{R}^{2d}$, $\phi = [\nabla f(x)^\top,\nabla f(x)^\top]^\top \in \mathbb{R}^{2d}$. Notice that $\zeta$ still follows an isotropic multivariate Gaussian distribution, we can simplify the integral in LHS of (32) as:

$$\int_{\mathcal{S}_\zeta(\phi)}\langle\phi,\zeta\rangle\,p(\zeta)d\zeta \tag{104}$$

where $\mathcal{S}_\zeta(\phi) = \left\{\zeta\mid\left\|\zeta-\frac{1}{\mu L}\phi\right\|_2^2 < \frac{\|\phi\|_2^2}{\mu^2 L^2}\right\}$.

We argue that for any rotation matrix $R \in \mathbb{R}^{2d\times 2d}$, i.e., $\det(R) = 1$ and $R^\top = R^{-1}$. We have

$$\int_{\mathcal{S}_\zeta(\phi)}\langle\phi,\zeta\rangle\,p(\zeta)d\zeta = \int_{\mathcal{S}_\zeta(R\phi)}\langle R\phi,\zeta\rangle\,p(\zeta)d\zeta. \tag{105}$$

To see that, we can rotate $\zeta$ by $R$. Denote $\zeta' = R^\top\zeta$, we first have

$$\mathcal{S}_\zeta(R\phi) = \left\{ \zeta \mid \left\| \zeta - \frac{1}{\mu L} R\phi \right\|_2^2 < \frac{\|\phi\|_2^2}{\mu^2 L^2} \right\} = \left\{ R\zeta' \mid \left\| \zeta' - \frac{1}{\mu L} \phi \right\|_2^2 < \frac{\|\phi\|_2^2}{\mu^2 L^2} \right\} = \{R\zeta' \mid \zeta' \in \mathcal{S}_{\zeta'}(\phi)\}$$

(106)

$$\int_{\mathcal{S}_\zeta(R\phi)} \langle R\phi, \zeta \rangle \, p(\zeta) d\zeta = \int_{\{R\zeta' \mid \zeta' \in \mathcal{S}_{\zeta'}(\phi)\}} \langle R\phi, R\zeta' \rangle \, p(R\zeta') dR\zeta' = \int_{\mathcal{S}_{\zeta'}(\phi)} \langle \phi, \zeta' \rangle \, p(\zeta') d\zeta',$$

(107)

where we use the property of $p(\cdot)$: $p(R\zeta') = p(\zeta')$.

Now we denote $\phi' = [\|\phi\|, 0, ..., 0]^\top \in \mathbb{R}^{2d}$, it is easy to see that $\phi'$ is a rotated version of $\phi$, i.e., there exists a rotation matrix $R'$ such that $\phi' = R'\phi$. Denote $\zeta = [\zeta_1, ..., \zeta_{2d}]^\top$, and $\zeta_{/1} = [\zeta_2, ..., \zeta_{2d}]^\top$. We have

$$\int_{\mathcal{S}_\zeta(\phi)} \langle \phi, \zeta \rangle \, p(\zeta) d\zeta$$

(108)

$$= \int_{\mathcal{S}_\zeta(\phi')} \langle \phi', \zeta \rangle \, p(\zeta) d\zeta$$

(109)

$$= \|\phi\| \int_{\left(\zeta_1 - \frac{\|\phi\|}{\mu L}\right)^2 + \zeta_2^2 + ... + \zeta_{2d}^2 \le \frac{\|\phi\|^2}{\mu^2 L^2}} \zeta_1 p(\zeta) d\zeta$$

(110)

$$= \|\phi\| \int_0^{\frac{2\|\phi\|}{\mu L}} \zeta_1 \left( \int_{\zeta_2^2 + ... + \zeta_{2d}^2 \le \frac{\|\phi\|^2}{\mu^2 L^2} - \left(\zeta_1 - \frac{\|\phi\|}{\mu L}\right)^2} p(\zeta_{/1}) d\zeta_{/1} \right) p(\zeta_1) d\zeta_1.$$

(111)

Notice that $\zeta_2, ..., \zeta_{2d}$ are i.i.d and following standard Gaussian distribution, and hence $\zeta_2^2 + ... + \zeta_{2d}^2$ follows the Chi-square distribution with $2d - 1$ degrees of freedom. Therefore,

$$\int_{\mathcal{S}_\zeta(\phi)} \langle \phi, \zeta \rangle \, p(\zeta) d\zeta$$

(112)

$$= \|\phi\| \int_0^{\frac{2\|\phi\|}{\mu L}} \zeta_1 F_{2d-1} \left( \frac{\|\phi\|^2}{\mu^2 L^2} - \left( \zeta_1 - \frac{\|\phi\|}{\mu L} \right)^2 \right) p(\zeta_1) d\zeta_1$$

(113)

$$= \|\phi\| \int_0^{\frac{2\|\phi\|}{\mu L}} \zeta_1 F_{2d-1} \left( \left( \frac{2\|\phi\|}{\mu L} - \zeta_1 \right) \zeta_1 \right) p(\zeta_1) d\zeta_1$$

(114)

$$= \sqrt{2} \|\nabla f(x)\| \int_0^{\frac{2\sqrt{2}\|\nabla f(x)\|}{\mu L}} \zeta_1 F_{2d-1} \left( \left( \frac{2\sqrt{2}\|\nabla f(x)\|}{\mu L} - \zeta_1 \right) \zeta_1 \right) p(\zeta_1) d\zeta_1$$

(115)

$$= h(\|\nabla f(x)\|, \mu L, d).$$

(116)

$\square$

*Proof of Lemma 6.* We define the fucntion $q(u, d) : \mathbb{R}_+ \times \mathbb{Z}_+ \to \mathbb{R}_+$ as follows:

$$q(u, d) = \int_0^{2\sqrt{2}u} x F_{2d-1} \left( \left( 2\sqrt{2}u - x \right) x \right) p(x) dx.$$

(117)

Notice that $h(v, r, d) = \sqrt{2} v q(v/r, d)$.

We first need to prove an important property of the function $q(u, d)$:

$$\lim_{u \to \infty} q(u, d) = \frac{1}{\sqrt{2\pi}}.$$

Consider an arbitrary $\epsilon > 0$. Since $\int_0^{+\infty} xp(x)dx = \frac{1}{\sqrt{2\pi}}$, there exists $N_2 > N_1 > 0$ and such that

$$0 < \int_0^{N_1} xp(x)dx \le \frac{\epsilon}{3}, \tag{118}$$

$$0 < \int_{N_2}^{\infty} xp(x)dx \le \frac{\epsilon}{3}. \tag{119}$$

On the other hands, for every $u > \frac{N_2}{\sqrt{2}}$, since $\left(2\sqrt{2}u - x\right)x$ is monotonically increasing on $[N_1, N_2]$, we have

$$\int_{N_1}^{N_2} xF_{2d-1}\left(\left(2\sqrt{2}u - x\right)x\right)p(x)dx > F_{2d-1}\left(\left(2\sqrt{2}u - N_1\right)N_1\right)\int_{N_1}^{N_2} xp(x)dx. \tag{120}$$

Notice that

$$\lim_{u \to \infty} F_{2d-1}\left(\left(2\sqrt{2}u - N_1\right)N_1\right) = 1,$$

there must exist a number $N_3$ such that if $u > N_3$, then

$$F_{2d-1}\left(\left(2\sqrt{2}u - N_1\right)N_1\right) > 1 - \sqrt{2\pi}\epsilon. \tag{121}$$

Putting together (120) and (121), because $0 \le F_{2d-1}\left(\left(2\sqrt{2}u - x\right)x\right) \le 1$, if $u > \max\{\frac{N_2}{\sqrt{2}}, N_3\}$, we can obtain

$$0 < \int_0^{+\infty} xp(x)dx - \int_0^{2\sqrt{2}u} xF_{2d-1}\left(\left(2\sqrt{2}u - x\right)x\right)p(x)dx \tag{122}$$

$$\le \frac{2\epsilon}{3} + \int_{N_1}^{N_2} xp(x)dx - \int_{N_1}^{N_2} xF_{2d-1}\left(\left(2\sqrt{2}u - x\right)x\right)p(x)dx \tag{123}$$

$$\le \frac{2\epsilon}{3} + \int_{N_1}^{N_2} xp(x)dx - F_{2d-1}\left(\left(2\sqrt{2}u - N_1\right)N_1\right)\int_{N_1}^{N_2} xp(x)dx \tag{124}$$

$$\le \frac{2\epsilon}{3} + \int_{N_1}^{N_2} xp(x)dx - \left(1 - \sqrt{2\pi}\epsilon\right)\int_{N_1}^{N_2} xp(x)dx \tag{125}$$

$$= \frac{2\epsilon}{3} + \frac{\epsilon}{3}\int_{N_1}^{N_2} xp(x)dx < \frac{2\epsilon}{3} + \sqrt{2\pi}\epsilon\frac{1}{\sqrt{2\pi}} = \epsilon. \tag{126}$$

Taking $\epsilon \to 0$, hence we know that

$$\lim_{u \to \infty} q(u, d) = \int_0^{+\infty} xp(x)dx = \frac{1}{\sqrt{2\pi}}.$$

Since $\lim_{u \to \infty} q(u, d) = \frac{1}{\sqrt{2\pi}}$, there exists a constant $C_d$ such that whenever $\left(\frac{1}{2\sqrt{\pi}} + \frac{1}{4}\right)u > \frac{1}{4}C_d$, we have

$$q(u, d) \ge \frac{1}{\sqrt{2\pi}} - \left(\frac{1}{2\sqrt{2\pi}} - \frac{1}{4\sqrt{2}}\right) = \frac{1}{2\sqrt{2\pi}} + \frac{1}{4\sqrt{2}}. \tag{127}$$

Therefore, whenever $\left(\frac{1}{2\sqrt{\pi}} + \frac{1}{4}\right)v > \frac{1}{4}C_d r$, we have

$$h(v, r, d) = \sqrt{2}vq(v/r, d) \ge \left(\frac{1}{2\sqrt{\pi}} + \frac{1}{4}\right)v \ge \left(\frac{1}{2\sqrt{\pi}} + \frac{1}{4}\right)v - \frac{1}{4}C_d r. \tag{128}$$

On the other hand, when $\left(\frac{1}{2\sqrt{\pi}} + \frac{1}{4}\right)v \le \frac{1}{4}C_d r$, we have

$$h(v, r, d) = \sqrt{2}vq(v/r, d) \ge 0 \ge \left(\frac{1}{2\sqrt{\pi}} + \frac{1}{4}\right)v - \frac{1}{4}C_d r. \tag{129}$$

$\square$

## C EXPERIMENT DETAILS

### C.1 HYPERPARAMETER CHOICES FOR THE EXPERIMENTS IN SECTION 4.1

Figure 8 and 9 show the performance of tested algorithms in Figure 3 under different hyperparameter settings. For gradient-based algorithms, ZO-SGD, SCOBO, and ZO-RankSGD, we tune the stepsize and set $\gamma = 0.1$ for the line search. We need to remark that when implementing the SCOBO (Cai et al., 2022), we remove the sparsity constraint because we found that it will lead to degraded performance for non-sparse problems like the ones we tested. For GLD-Fast, we tune for the diameter of search sparse, denoted as $\mu$. For CMA-ES, we tune for the initial variance, also denoted as $\mu$ in the figures. To run the experiment in Figure 3, we select the optimal choices of hyperparameters based on Figure 8 and 9 for each algorithm, respectively.

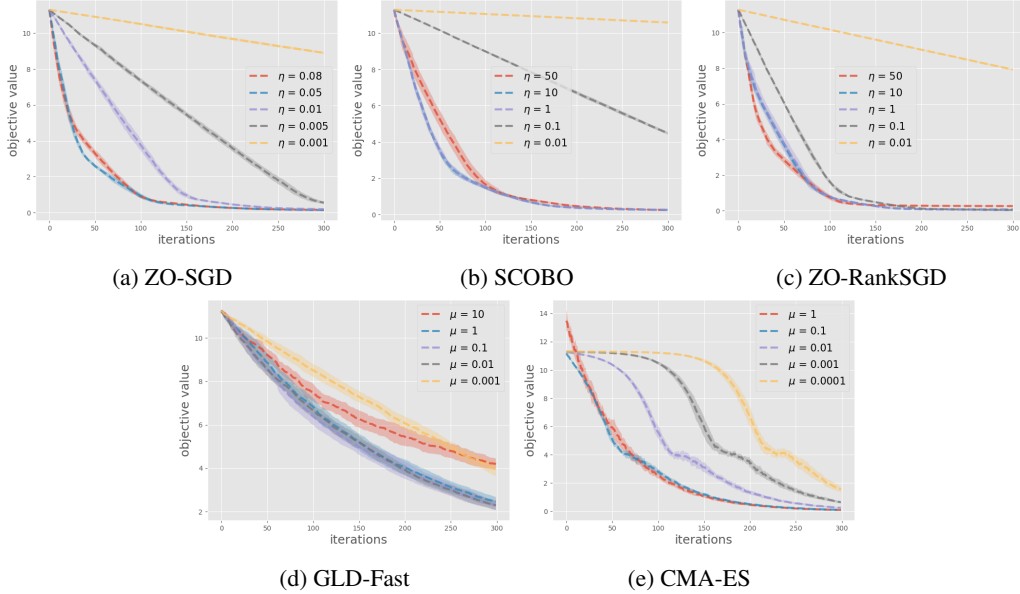

Figure 8: Hyperparameter tuning on Quadratic function.

### C.2 EXPERIMENTS ON HIGH-DIMENSIONAL OPTIMIZATION PROBLEM

In this section, we examine the performance of ZO-RankSGD on a high-dimensional optimization problem, and the results is presented in Figure 10. Specifically, we adopt the same setting as the experiments in Figure 3, except that we increase the problem dimension to 10000. It is worth noting that the CMA-ES is no longer included in this experiment due to the overwhelming computation time. This is mainly because, at this problem scale, CMA-ES requires updating a $10000 \times 10000$ covariance matrix.

As we can see, the phenomenon is similar to the results presented in Figure 3, showcasing the ability of ZO-RankSGD on tackling high-dimensional problems.

We also provide more details for the hyperparameters selection in these experiments as we did in Section C.1. See Figure 11 and 12 for this information.

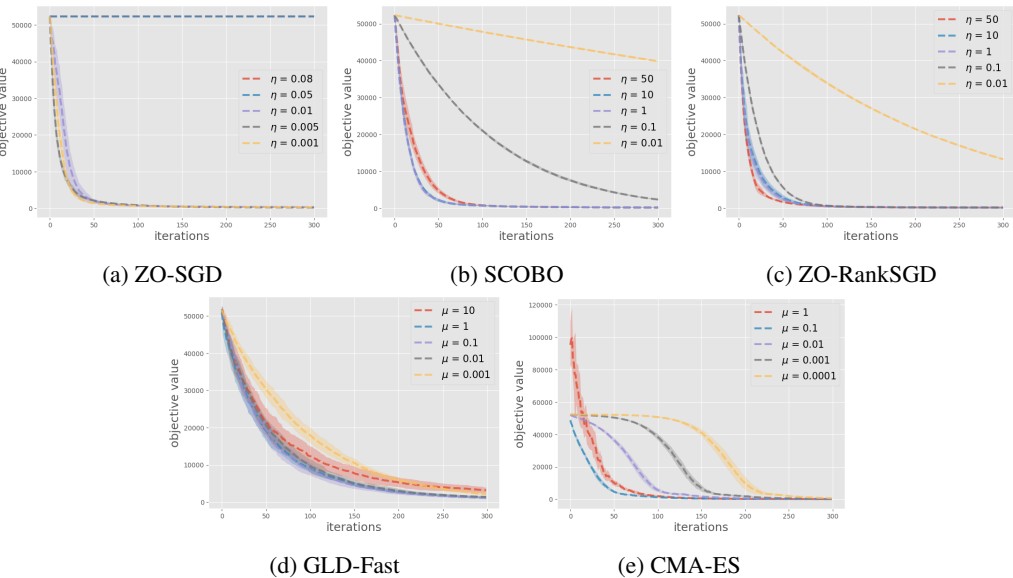

Figure 9: Hyperparameter tuning on Rosenbrock function.

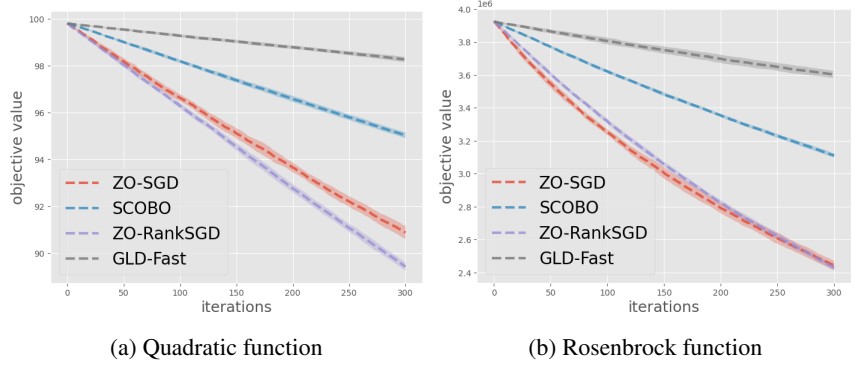

Figure 10: Performance of different algorithms on the 10000-dims optimization problems.

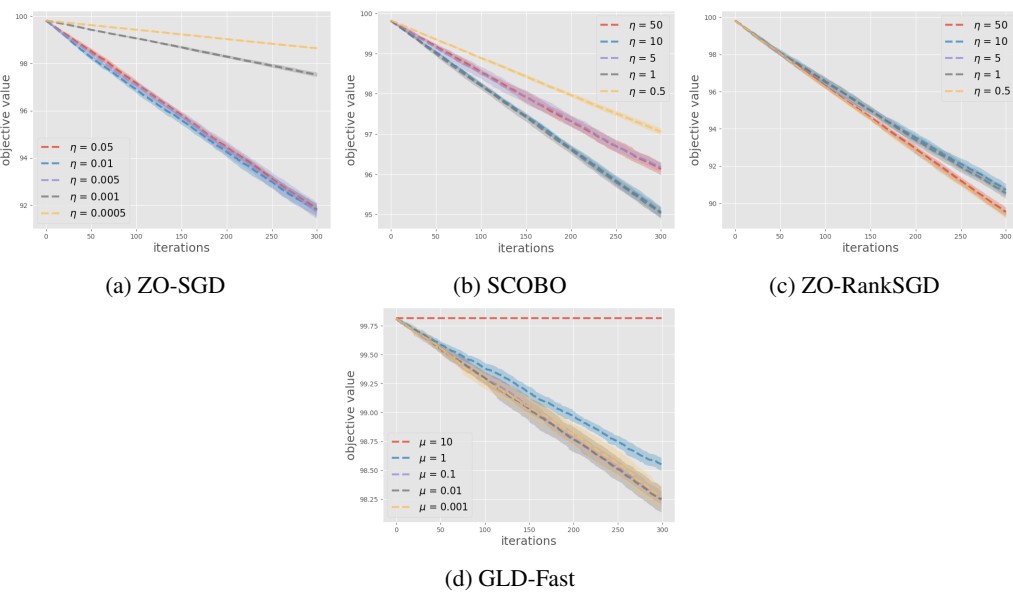

Figure 11: Hyperparameter tuning on Quadratic function.

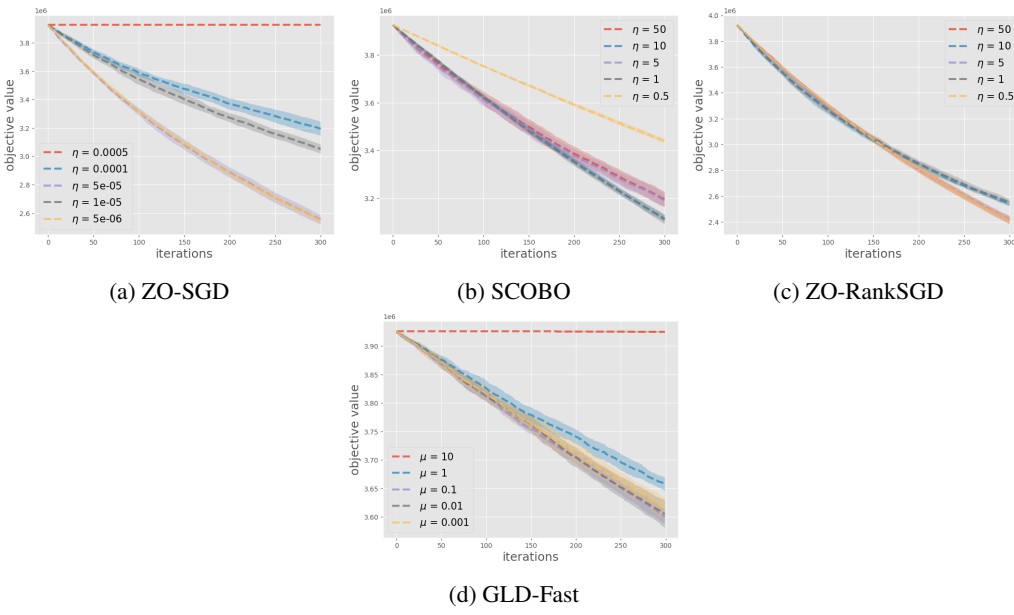

Figure 12: Hyperparameter tuning on Rosenbrock function.

## C.3 EXPERIMENTS WITH NOISY RANKING ORACLES

In this section, we present preliminary results to assess the performance of ZO-RankSGD when confronted with noisy ranking oracles. It is essential to note that, unlike the comparison oracle introduced by Cai et al. (2022), which employs flipping probabilities to represent errors in noisy comparison feedback, modeling errors in noisy ranking feedback is not straightforward.

To simulate noisy ranking oracles for our experiments, we empirically introduce Gaussian noise to the ground-truth function value. We then construct the corresponding noisy ranking feedback based on the perturbed values.

Figure 13 illustrates the performance of both ZO-SGD and ZO-RankSGD under varying levels of noise, denoted by the variance parameter $\eta$, added to the function value. We select the optimal stepsize independently for ZO-SGD and ZO-RankSGD. Remarkably, ZO-RankSGD demonstrates resilience to additive noise across different levels of variance, consistently maintaining performance comparable to ZO-SGD. Notably, for the Rosenbrock function, ZO-RankSGD outperforms ZO-SGD, indicating superior robustness to additive noise. We speculate that this advantage stems from ZO-RankSGD relying solely on rank information for optimization, which may exhibit less variability under mild additive noise.

Looking forward, our aim is to establish a well-defined framework for noisy ranking oracles and extend the theoretical analysis of ZO-RankSGD within this context. Additionally, we hope to explore the theoretical robustness of ZO-RankSGD to noise.

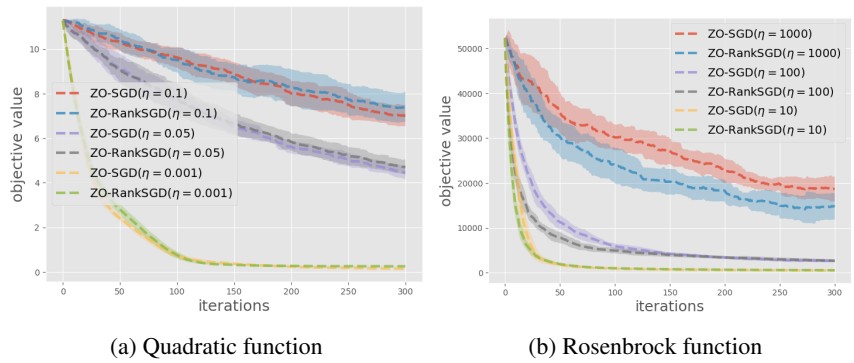

(a) Quadratic function          (b) Rosenbrock function

Figure 13: Performance of ZO-RankSGD and ZO-SGD under noisy feedback.

## C.4 DETAILS FOR THE EXPERIMENTS IN SECTION 4.2

**Problem dimension.** In this experiment, for all three scenarios from Mujoco environment, we seek to optimize a linear policy mapping states to actions. Specifically, the dimensions for the Reacher-v2, Swimmer-v2, and HalfCheetah-v2 are 24, 18, 108 respectively.

**Comparing ZO-RankSGD with SCOBO in policy optimization.** In this part, we delve into a detailed comparison between ZO-RankSGD and SCOBO. It is important to note that a direct comparison is challenging, as they depend on fundamentally different query oracles. However, we propose an alternative comparison approach from an information perspective. Specifically, given a budget of 5 query points per iteration, SCOBO can make only 4 independent pairwise comparisons, while ZO-RankSGD can obtain information from 10 dependent pairwise comparisons by querying a $(5,5)$-ranking oracle.

From this standpoint, we anticipate that ZO-RankSGD would outshine SCOBO with $m = 5$ (which can only query information of 5 points via 4 independent pairwise comparisons), but might fall short when compared to SCOBO with $m = 11$ (which can query information of 11 points via 10 independent pairwise comparisons).

To test this hypothesis, we benchmark ZO-RankSGD, SCOBO ($m = 5$), and SCOBO ($m = 11$) on the same policy optimization problem discussed in Section 4.2. The results, shown in Figure 14, align precisely with our prediction, thus validating our perspective.

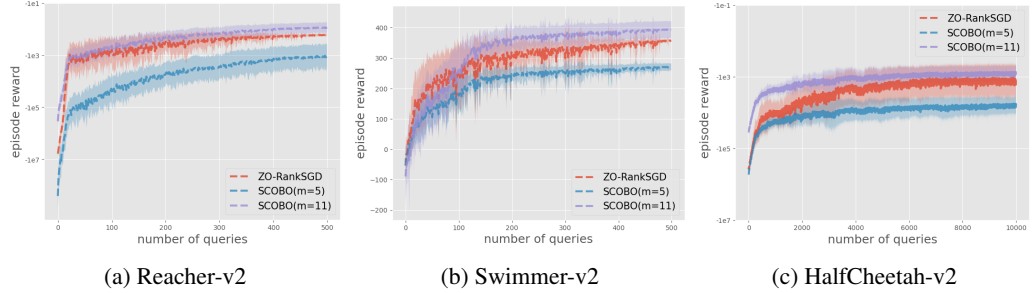

|  |  |  |
|:---:|:---:|:---:|
| (a) Reacher-v2 | (b) Swimmer-v2 | (c) HalfCheetah-v2 |

Figure 14: Perfomance of ZO-RankSGD and SCOBO on three MuJoCo environments

### C.5 DETAILS FOR THE EXPERIMENT IN SECTION 4.3

**Modified ZO-RankSGD algorithm for optimizing latent embeddings of Stable Diffusion.** To enhance the efficiency of Algorithm 1, we make a modification to preserve the best image obtained during the optimization process. Specifically, in the original algorithm, the best point among all queried images is not saved, which can lead to inefficiencies. Therefore, we modify the algorithm to store the best point in the gradient estimation step as $x^{**}$ and add it to the later line search step. This modification can be viewed as a combination of ZO-RankSGD and Direct Search (Powell, 1998). Another useful feature of Algorithm 3 is that if the best point is not updated in the line search step, the algorithm returns to the gradient estimation step to form a more accurate gradient estimator. The modified algorithm is presented in Algorithm 3. At every iteration in Algorithm 3, we evaluate the latent embeddings by passing them to the DPM-solver with Stable Diffusion and then ask human or CLIP model to rank the generated images.

---

**Algorithm 3** Modified ZO-RankSGD algorithm for optimizing latent embeddings of Stable Diffusion.

---

**Require:** Objective function $f$ (Evaluated by human or CLIP model), initial point $x_0$, number of queries $m$, stepsize $\eta$, smoothing parameter $\mu$, shrinking rate $\gamma \in (0, 1)$, number of trials $l$.
 1: Initialize the best point $x^* = x_0$.
 2: Initialize the gradient memory $\bar{g}$ with all-zero vectors.
 3: Set $\tau = 0$.
 4: **while** not terminated by user **do**
 5:      Sample $m$ i.i.d. direction $\{\xi_1, \cdots, \xi_m\}$ from $N(0, I)$.
 6:      Query $O_f^{(m,k)}$ with input $\mathcal{X}_1 = \{x^* + \mu\xi_1, \cdots, x^* + \mu\xi_m\}$ for some $k \leq m$. Denote $\mathbb{I}_1$ as the output.
 7:      Set $x^{**}$ to be the point in $\mathcal{X}_1$ with minimal objective value.
 8:      Compuate the gradient $\hat{g}$ using the ranking information $\mathbb{I}_1$ as in Algorithm 1.
 9:      $\bar{g} = (\tau\bar{g} + \hat{g})/(\tau + 1)$
10:      $\tau = \tau + 1$
11:      Query $O_f^{(m,1)}$ with input $\mathcal{X}_2 = \{x^*, x^{**}, x^* - \eta\bar{g}, x^* - \eta\gamma\bar{g}, ..., x^* - \eta\gamma^{m-2}\bar{g}\}$. Denote $\mathbb{I}_2$ as the output.
12:      **if** $1 \in \mathbb{I}_2$, i.e., $x^*$ has the minimal objective value **then**
13:          Go back to line 5.
14:      **else**
15:          Set $x^*$ to be the point in $\mathcal{X}_2$ with minimal objective value.
16:          Initialize the gradient memory $\bar{g}$ with all-zero vectors.
17:          Set $\tau = 0$.
18:      **end if**
19: **end while**

---

**The User Interface for Algorithm 3.** Figure 15 presents the corresponding user interface (UI) designed for collecting human feedback in Algorithm 3, where 6 images are presented to the users at each round. When the user receives the instruction "Please rank the following image from best to worst," it indicates that the algorithm is in the gradient estimation step. In this case, users are required to rank $k$ best images, where $k$ can be any number they choose. Then, the user receives the instruction "Please input the ID of the best image," indicating that the algorithm has moved to the line search step, and users only need to choose the best image from the presented images. This

interface enables easy and intuitive communication between the user and the algorithm, facilitating the optimization process.

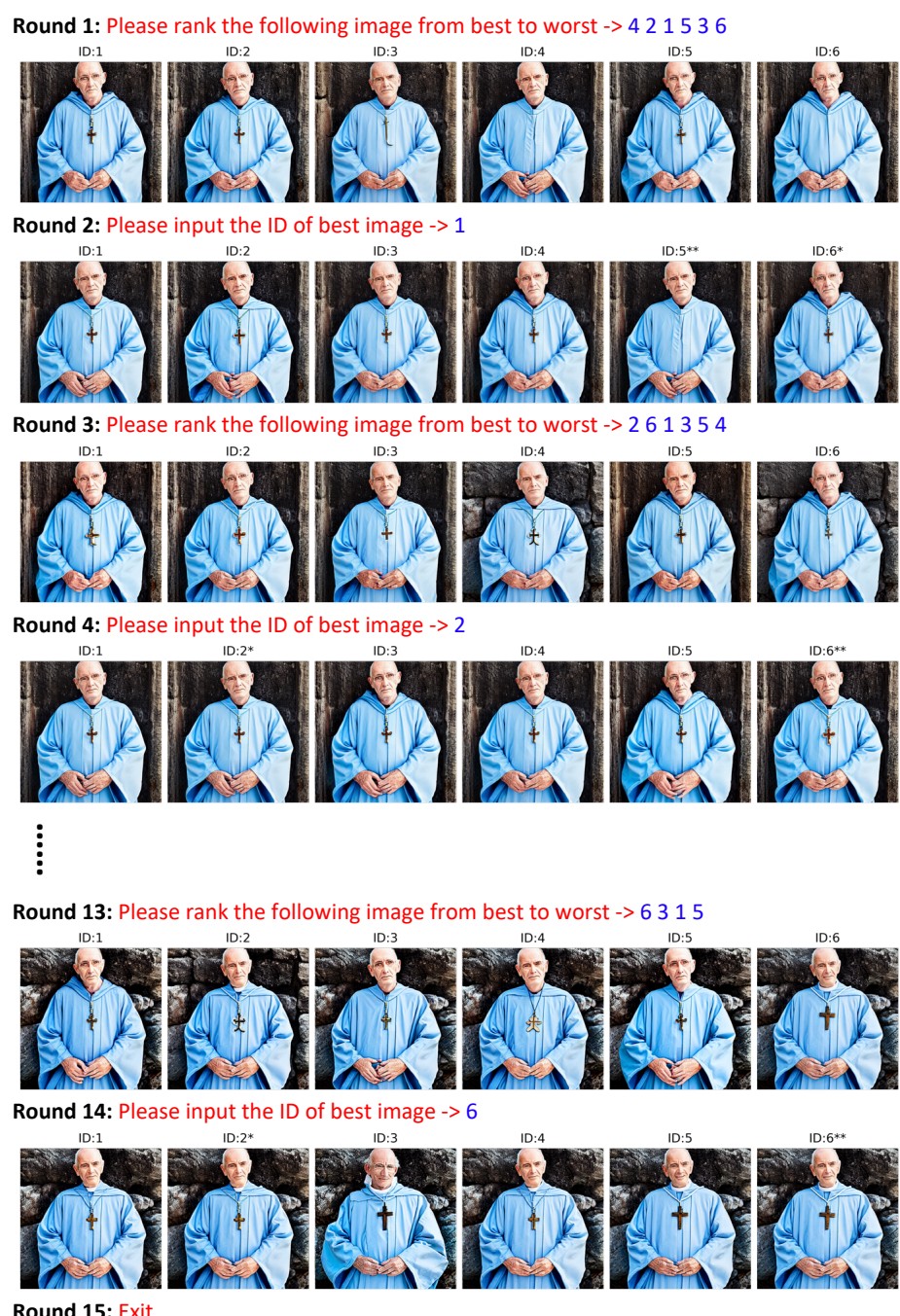

Figure 15: The User Interface of Algorithm 3.

In this experiment, we use some popular text prompts from the internet[1]. More examples like the ones in Figure 7 are presented in Figure 16.

---

[1] https://mpost.io/best-100-stable-diffusion-prompts-the-most-beautiful-ai-text-to-image-pro

**Other details.**     For all the examples in Figure 7 and Figure 16, we set the number of rounds for human feedback between 10 and 20, which was determined based on our experience with the optimization process. For the images obtained from the CLIP similarity score, we fixed the number of querying rounds to 50. Both the optimization from human feedback and CLIP similarity score used the same parameters for Algorithm 3: $\eta = 1$, $\mu = 0.1$, and $\gamma = 0.5$. Especially, the $\mu = 0.1$ is chosen according to the rule discussed in Section 3.1, as we select a sufficiently small $\mu$ value which still allows humans to perceive differences between perturbed images. Since the latent embedding of Stable Diffusion is $64 \times 3 \times 3$, the problem dimension of the optimization problem is 576.

| Prompt | Initial | Human | CLIP |

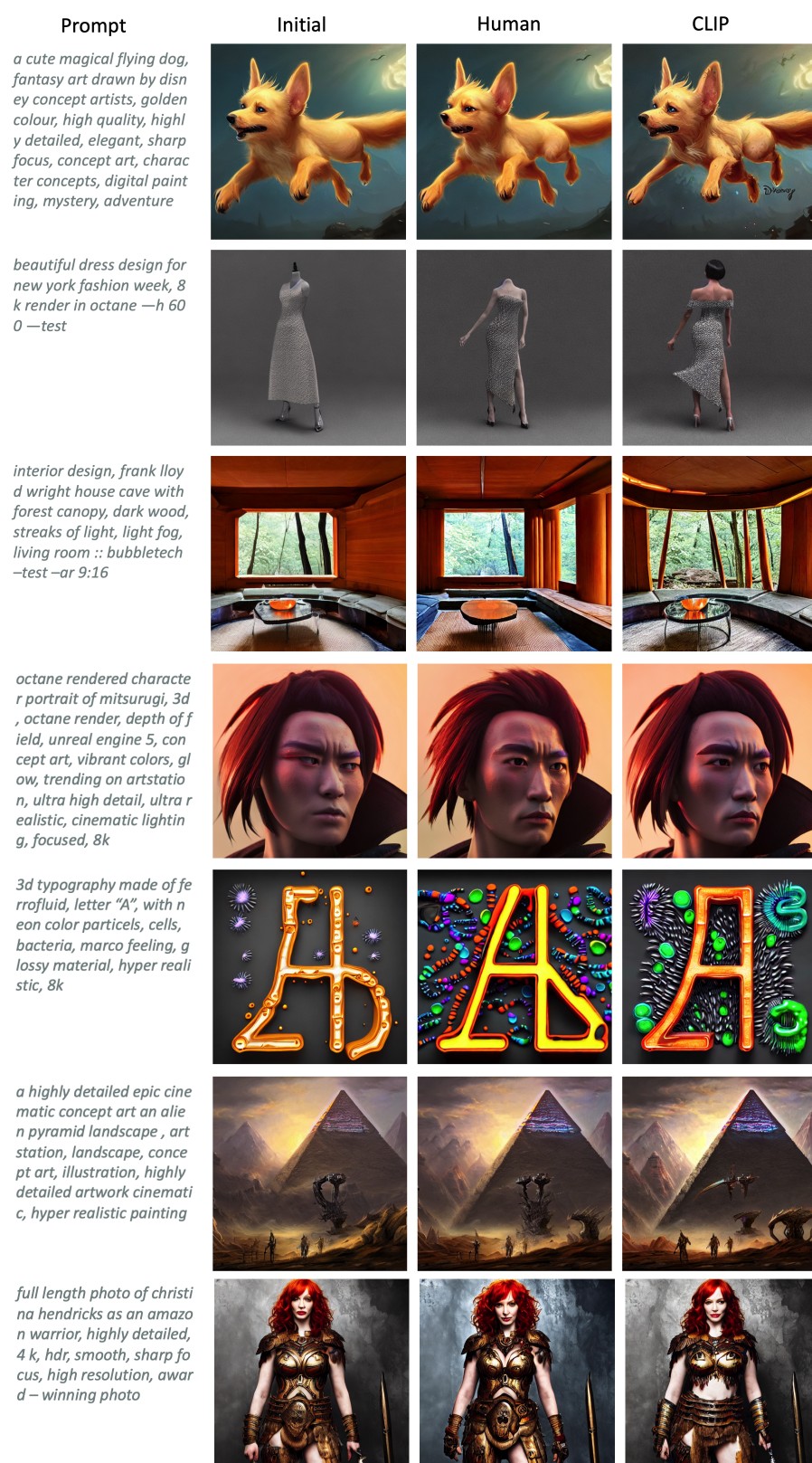

*a cute magical flying dog, fantasy art drawn by disn ey concept artists, golden colour, high quality, highl y detailed, elegant, sharp focus, concept art, charac ter concepts, digital paint ing, mystery, adventure*

*beautiful dress design for new york fashion week, 8 k render in octane —h 60 0 —test*

*interior design, frank lloy d wright house cave with forest canopy, dark wood, streaks of light, light fog, living room :: bubbletech —test –ar 9:16*

*octane rendered characte r portrait of mitsurugi, 3d , octane render, depth of f ield, unreal engine 5, con cept art, vibrant colors, gl ow, trending on artstatio n, ultra high detail, ultra r ealistic, cinematic lightin g, focused, 8k*

*3d typography made of fe rrofluid, letter "A", with n eon color particels, cells, bacteria, marco feeling, g lossy material, hyper reali stic, 8k*

*a highly detailed epic cine matic concept art an alie n pyramid landscape , art station, landscape, conce pt art, illustration, highly detailed artwork cinemati c, hyper realistic painting*

*full length photo of christi na hendricks as an amazo n warrior, highly detailed, 4 k, hdr, smooth, sharp fo cus, high resolution, awar d – winning photo*

Figure 16: More examples.

