# OpenReview forum: "Zeroth-Order Optimization Meets Human Feedback: Provable Learning via Ranking Oracles"
_ICLR.cc/2024/Conference — ICLR 2024 poster_

### Official Review · Reviewer_6pMr · 2023-10-16

**Soundness:** 3 good
**Presentation:** 2 fair
**Contribution:** 3 good
**Rating:** 6
**Confidence:** 3

**Summary:**

In this paper, the authors consider the problem of optimizing a black box function with only ranking feedback. The authors propose a novel estimator for gradient descent direction based on ranking information. They show that this estimator guarantees convergence to a stationary point. Empirical evaluations covering a diverse set of applications, including reinforcement learning and diffusion generative models, show the effectiveness of the proposed method.

**Strengths:**

1. As far as I know, this is the first algorithm utilizing ranking feedback that comes with a convergence guarantee. I briefly checked the proofs and found the technique establishing the main theoretical result (lemma 1) to be interesting.

2. The diversity of applications demonstrates the general applicability of the proposed algorithm. The image generation example is timely with the increasing popularity of aligning generative models with human preferences.

**Weaknesses:**

1. While the experiments are diverse, their comprehensiveness could be improved. The image generation experiment is very limited. While the three examples presented in the main paper exhibit some improvements over the baseline model, looking at additional examples in the appendix the improvements are harder to assess. Typically for image generation, some kind of human evaluation is necessary to provide a more objective evaluation.

2. Similarly, the reinforcement learning experiment compares only with a single baseline method (CMA-ES), which is not designed for RL. I think comparing with more RL-focused baselines such as [1] and [2] is more appropriate.

[1]: Lee, Kimin, Laura M. Smith, and Pieter Abbeel. "PEBBLE: Feedback-Efficient Interactive Reinforcement Learning via Relabeling Experience and Unsupervised Pre-training." International Conference on Machine Learning. PMLR, 2021.

[2]: Christiano, Paul F., et al. "Deep reinforcement learning from human preferences." Advances in neural information processing systems 30 (2017).

3. There are some details missing for the empirical evaluations. For example, in section 4.2, how does a policy generate multiple episodes for querying the ranking oracle? What is the network architecture to represent a policy?

**Questions:**

1. Towards the end of section 3.1 on how to choose $\mu$ in Algorithm 1, can the authors expand on how to choose $\mu$ in practice, for example in the stable diffusion experiment?

2. It is not clear how the line search budget is accounted for in Section 4.1. Is it the case that, with $m=15$ queries, gradient-based algorithms will use 10 queries to estimate $g_t$ and the final 5 queries are used for the line search?

3. Section 4.1 on investigating the impact of $m$ and $k$: Equation (10) seems to be dominated by $M_1(f, \mu)$ when $k$ and $m$ are of relatively large values. That means that the second variance is actually not the dominant quantity. Please let me know if my understanding is correct.

4. In section 4.2, the authors should compare with recent preference-based RL algorithms mentioned above. Additionally, it would be useful to include achieved rewards for standard RL algorithms with scalar rewards to show the performance gap (if any) using preference-based rewards.

---

> ### Author Response · Authors · 2023-11-14
> **Rebuttal reponse to reviewer 6pMr (1/2)**
>
> Thank you for inputting valuable feedback for this manuscript. Below is our response to your major concern.
>
> ### 1. The first concern is "The image generation experiment is very limited. While the three examples presented in the main paper exhibit some improvements over the baseline model, looking at additional examples in the appendix the improvements are harder to assess. Typically for image generation, some kind of human evaluation is necessary to provide a more objective evaluation."
>
> We acknowledge that it is indeed better to include an objective evaluation of the generated images. Regrettably, since the generation process for a single image requires a real human to provide 10-20 rounds of ranking feedback,  it is neither affordable for us to generate a sufficient amount of images using ZO-RankSGD for a rigorous evaluation nor to hire a real human for evaluation. Therefore, we wish to remain the main scope of this work within providing the first theoretical and principled framework for optimization with ranking oracles, and to inspire future research on incorporating this framework with more engineering consideration.
>
> ---
>
>
>
> ### 2. The second concern is "Similarly, the reinforcement learning experiment compares only with a single baseline method (CMA-ES), which is not designed for RL. I think comparing with more RL-focused baselines such as [1] and [2] is more appropriate."
>
> We agree with the reviewer that the baselines [1,2] are indeed closely related to the experiment setting in section 4.2, which represents a model-based RL approach for incorporating ranking feedback. However, we chose to not include them in this manuscript for the reason explained below.
>
> There are several major differences between the black-box optimization algorithms like CMA-ES and Zo-RankSGD and the model-based RL approach like the ones in [1,2], which make it very tricky to directly compare them experimentally.
>
> Specifically,  as illustrated by the previous works ”Evolution Strategies as a Scalable Alternative to Reinforcement Learning“ Salimans et. al and "Simple random search of static linear policies is competitive for reinforcement learning" Mania et. al, the black optimization algorithm like CMA-ES aims to offer model-free and off-the-shelf policy optimization without requiring models of the system dynamics, and it has several advantages over model-based RL algorithm: It enjoys extremely high parallelism when there are a sufficient amount of CPUs and is also invariant to action frequency and delayed rewards, tolerant of extremely long horizons, and does not need temporal discounting or value function approximation.
>
> From our perspective, the key difference is that the model-based RL approaches in [1,2] need to first learn a reward model after collecting a sufficient amount of human feedback before any policy optimization can happen, and, as reported in [1,2], there is usually an extra burden to leverage a lot of tricks to prevent the overfitting of such reward model. On the sharp contrary, model-free algorithms like CMA-ES and ZO-RankSGD start optimizing the policy once it receives ranking feedback.
>
> Given these differences, it is not a trivial task to directly compare a model-based RL algorithm and a model-free RL algorithm as it should cover many different aspects: Ease of hyperparameter tuning, parallel implementation, sample efficiency, etc.
>
> Therefore, we believe that this comparison is beyond the main scope of this work and hence would like to maintain the comparison within the context of black-box and model-free optimization algorithms. We hope this explanation can help the reviewer understand our motivation for such a decision.
>
>
>
> ---
>
>
>
> ### 3. The third concern is "There are some details missing for the empirical evaluations. For example, in section 4.2, how does a policy generate multiple episodes for querying the ranking oracle? What is the network architecture to represent a policy?"
>
> Sorry for the confusion. Due to the page limit, we did not include many details in this manuscript and only mentioned that we adopted the same setting as Cai et al 2022. For the question "how does a policy generate multiple episodes for querying the ranking oracle?", when applying a typical black-box optimization algorithm for policy optimization, a general procedure is to first add small random perturbations to the policy to obtain multiple different candidate policies, and then the multiple episodes are generated from these candidate policy. As for the network architecture, we followed Cai et al 2022 to use a simple linear policy in our experiment.

---

> > ### Author Response · Authors · 2023-11-14
> > **Rebuttal reponse to reviewer 6pMr (2/2)**
> >
> > ### 4. Here are our answers to the four questions raised by the reviewer:
> >
> > * "Towards the end of section 3.1 on how to choose $\mu$ in Algorithm 1, can the authors expand on how to choose $\mu$ in practice, for example in the stable diffusion experiment?"
> >
> >   As explained in section 3.1, the principle is to choose the $\mu$ as small as possible while human is still able to perceive the difference between generated images. For the stable diffusion experiment, we just start from a large value of $\mu$, and then gradually decrease it until it becomes a bit hard to tell the difference of generated image.
> >
> > * "It is not clear how the line search budget is accounted for in Section 4.1. Is it the case that, with m=15 queries, gradient-based algorithms will use 10 queries to estimate $g_t$ and the final 5 queries are used for the line search?"
> >
> >   Yes, your understanding is correct.
> >
> > * "Section 4.1 on investigating the impact of m and k: Equation (10) seems to be dominated by $M_1(f,\mu)$ when $k$ and $m$ are of relatively large values. That means that the second variance is actually not the dominant quantity. Please let me know if my understanding is correct."
> >
> >   Yes, your understanding is correct. When $k$ and $m$ are sufficiently large, the only remaining variance term is $M_1(f,\mu)$.
> >
> > * "In section 4.2, the authors should compare with recent preference-based RL algorithms mentioned above. Additionally, it would be useful to include achieved rewards for standard RL algorithms with scalar rewards to show the performance gap (if any) using preference-based rewards."
> >
> >   As discussed in our last response, it is tricky to compare to model-based RL algorithms. However, comparing rank-based algorithms to value-based algorithms is indeed a valuable suggestion. Therefore, similar to our experiment in section 4.1, we also include the value-based zero-order algorithm for the RL experiment in the revised manuscript. The results are interesting, ZO-RankSGD exhibits performance on par with ZO-SGD, reinforcing our findings from the experiment illustrated in Figure 3 and underscoring the effectiveness of the ranking oracle in providing substantial optimization-relevant information.

---

> > > ### Comment · Reviewer_6pMr · 2023-11-22
> > > **Thanks for the response**
> > >
> > > I want to thank the authors for their detailed responses. However, my main concerns about the empirical evaluations still remain. I will main my score.

---

### Official Review · Reviewer_T8Lm · 2023-10-26

**Soundness:** 4 excellent
**Presentation:** 4 excellent
**Contribution:** 3 good
**Rating:** 6
**Confidence:** 4

**Summary:**

This paper studies the zeroth order optimization problem where the algorithm is only allowed to query a deterministic ranking oracle. That is, given two points $x_1,x_2$, the oracle returns the point at which the value of the objective function is smaller. Under smoothness assumptions, this paper designs the ZO-RankSGD algorithm that provably converges to a local with rate $\sqrt{d/T}$, where $d$ is the dimension. Empirically, this paper also shows that ZO-RankSGD performs competitively to ZO-SGD (where the algorithm can query the function value). It is also shown that ZO-RankSGD can be used to search over the random seeds in the diffusion model using human feedback.

**Strengths:**

-	This paper shows both theoretically and empirically that the proposed ZO-RankSGD algorithm has competitive performance.
-	The estimator for (m,k)-ranking oracle is neat and novel, and it extends the pairwise comparison oracle in classic RLHF framework. This extension is also well-motivated by the application with diffusion model, where it is reasonable and practical to generate more than two images per round.
-	This paper is well-written and easy-to-follow.

**Weaknesses:**

-	My main concern about this paper is that the oracles are deterministic. However, in practical settings, even human labelers have a lot of stochasticity/inconsistency when generating feedback (see e.g. [1]). Hence there is a gap between the theory and practice even though the algorithm performs well in the diffusion model application. Since most of the prior works assume a stochastic oracle, this setting needs to be further justified.
-	There is no rigorous quantitative results in real-world applications (i.e., the diffusion model). In the paper, there are only a few examples of images generated using ZO-RankSGD and human feedback.

[1] Dubois, Yann, et al. Alpacafarm: A simulation framework for methods that learn from human feedback.

**Questions:**

-	Is the convergence rate in Corollary 1 optimal in this setting?

---

> ### Author Response · Authors · 2023-11-14
> **Rebuttal response to reviewer T8Lm**
>
> Thank you for inputting valuable feedback for this manuscript. Below is our response to your major concern.
>
> ### 1. The first concern is "My main concern about this paper is that the oracles are deterministic. However, in practical settings, even human labelers have a lot of stochasticity/inconsistency when generating feedback (see e.g. [1]). Hence there is a gap between the theory and practice even though the algorithm performs well in the diffusion model application. Since most of the prior works assume a stochastic oracle, this setting needs to be further justified."
>
> We agree with you that the extension to noisy ranking oracles is indeed important, and this is why we mention that it is an important future direction in the previous manuscript. It is essential to note that, unlike the comparison oracle introduced by Cai et al. (2022), which employs flipping probabilities to represent errors in noisy comparison feedback, formulating the errors in noisy ranking feedback is not straightforward. This is the key reason why we did not consider including the results on noisy ranking feedback.
>
> As a response to this concern, we have added some preliminary experiments for investigating the performance of ZO-RankSGD on handling noisy ranking feedback, see Appendix C.3 of the revised manuscript for more details.
>
> Here we provide a summary for these new results:
>
> To simulate noisy ranking oracles for our preliminary experiments, we consider the scenario of directly adding Gaussian noise to the ground-truth function value, then construct the corresponding noisy ranking feedback based on the perturbed values.
>
> In these new experiments, we found that ZO-RankSGD demonstrates resilience to additive noise across different levels of variance, consistently maintaining performance comparable to ZO-SGD. Notably, for the Rosenbrock function, ZO-RankSGD outperforms ZO-SGD, indicating superior robustness to additive noise. We speculate that this advantage stems from ZO-RankSGD relying solely on rank information for optimization, which may exhibit less variability under mild additive noise.
>
> As one of the most important future directions, we hope to establish a well-defined formulation for the noisy ranking oracles and extend the theoretical analysis of ZO-RankSGD within this context.
>
> ---
>
> ### 2. The second concern is "There is no rigorous quantitative results in real-world applications (i.e., the diffusion model). In the paper, there are only a few examples of images generated using ZO-RankSGD and human feedback."
>
> We acknowledge that it is indeed better to include a quantitative evaluation of the generated images. Unfortunately, we were unable to do that for two reasons. Firstly, it is unaffordable for us to generate a sufficient amount of images using ZO-RankSGD for a rigorous evaluation, because the generation process for a single image requires a real human to provide 10-20 rounds of ranking feedback. Secondly,  there is still no widely acknowledged metric to access the images generated from Stable Diffusion. A common way in existing literature on diffusion models is to report metrics like FID for toy datasets, and then show a few examples for Stable Diffusion. In our work, FID is not a good metric as it does not reflect human preference.
>
> In all, the main scope of this work is only to provide the first theoretical and principled framework for optimization with ranking oracles, and to inspire future research on incorporating this framework with more engineering consideration.
>
> ---
>
> ### 3. One question is "Is the convergence rate in Corollary 1 optimal in this setting?"
>
> Yes, our convergence rate in Corollary 1 is optimal, according to the conclusion in "Optimal rates for zero-order convex optimization: the power of two function evaluations" Duchi et al.

---

> > ### Comment · Reviewer_T8Lm · 2023-11-20
> > **Follow-up questions**
> >
> > Thank you for the rebuttal. Could you specify how the noise is added to the ground-truth function for the new noisy experiments with Gaussian noise? Do you mean that every time the ranking oracle $S_f(x,\xi_1,\xi_2, \mu)$ is called, there is an additive Gaussian noise $\epsilon$ in the form $sign(f(x+\mu\xi_1)-f(x+\mu\xi_2)+\epsilon)$?

---

> > > ### Author Response · Authors · 2023-11-21
> > > **Reply for the question**
> > >
> > > $S_f(x,\xi_1,\xi_2,\mu)$ is the comparison operator, and we used it in paper only for expressing the formula more conveniently. In actual implementation, what we did is: Given input query $x_1$, ..., $x_m$, we first compute the function value $f(x_1),...,f(x_m)$, then add additive noise to the function value and return the sorting result, i.e., to return $\text{argsort}(f(x_1)+\xi_1,...,f(x_m)+\xi_m)$, $\xi_i\sim\mathcal{N}(0,\eta)$. Note that we add new noise to the ranking oracles every time, and this is to simulate the scenario where users may give different ranking feedback even for the same group of images.

---

### Official Review · Reviewer_BfNk · 2023-11-03

**Soundness:** 2 fair
**Presentation:** 3 good
**Contribution:** 2 fair
**Rating:** 6
**Confidence:** 4

**Summary:**

In this work, the authors introduces a novel optimization algorithm, called ZO-RankSGD, designed for solving optimization problems where only ranking oracles of the objective function are available. The authors demonstrated the effectiveness of ZO-RankSGD through both simulated and real-world tasks, such as image generation guided by human feedback. Additionally, they also explored the influence of various ranking oracles on optimization performance and offer recommendations for designing user interfaces that facilitate ranking feedback. The authors further propose future research directions that include adapting the algorithm to manage noisy and uncertain ranking feedbacks, integrating it with additional methods, and applying it to a broader range of cases beyond those with  human feedback.

**Strengths:**

Strengths:

- The writing of the paper is clear and easy to follow. In particular, its concise articulation of both the problem at hand and the proposed solution makes it much easier in understanding the paper. The authors also provided a rigorous mathematical formulation of the optimization challenge and provided the underlying intuition of their algorithm. Furthermore, they proved theoretical guarantess for the convergence of ZO-RankSGD and substantiate its practical effectiveness with experimental results across diverse settings.


- The second strength of the paper is its comprehensive experiments of how different ranking oracles affect the optimization results. The authors provide valuable insights into the design of user interfaces useful for ranking feedback and propose methods to enhance the query efficiency of the algorithm.

**Weaknesses:**

Weakness:


- Limited experments and unsupported claims: the evaluation of the algorithm's performance on noisy and uncertain ranking feedback is very limited. Also, the authors suggest furture direction for extending the algorithm to handle the aforementioned scenarios, however, they don't have any empirical results to support the claim. I would suggest the authors to provide at least some preliminary results for coroborate the argument.

- Narrow focus: the authors only focused on image generation with human feedback, which is also a bit limited. For human feedbacks, it's more commonly used in tuning language models, though there are recent works start to explore how can we incoporate human feedbacks into improving diffusion models. The method itself is interesting, and it would be a great add if the authors can show that the method is also effective for tuning language models.


---
Post rebuttal: thanks to the authors response. I think it has addressed most of my concerns. I will increase the score accordingly.

**Questions:**

see above.

**Details Of Ethics Concerns:**

see weakness.

---

> ### Author Response · Authors · 2023-11-14
> **Rebuttal response to reviewer BfNk**
>
> Thank you for inputting valuable feedback for this manuscript. Below is our response to your major concern.
>
> ### 1. The first concern is "the evaluation of the algorithm's performance on noisy and uncertain ranking feedback is very limited ..., they don't have any empirical results to support the claim. I would suggest the authors to provide at least some preliminary results for coroborate the argument."
>
>
>    As a response to this concern, we have added some preliminary experiments for investigating the performance of ZO-RankSGD on handling noisy ranking feedback, see Appendix C.3 of the revised manuscript for more details.
>
>    Here we provide a summary for these new results:
>
>    First of all, It is essential to note that, unlike the comparison oracle introduced by Cai et al. (2022), which employs flipping probabilities to represent errors in noisy comparison feedback, formulating the errors in noisy ranking feedback is not straightforward. This is the key reason why we did not consider including the results on noisy ranking feedback.
>
>    To simulate noisy ranking oracles for our preliminary experiments, we consider the scenario of directly adding Gaussian noise to the ground-truth function value, then construct the corresponding noisy ranking feedback based on the perturbed values.
>
>    In these new experiments, we found that ZO-RankSGD demonstrates resilience to additive noise across different levels of variance, consistently maintaining performance comparable to ZO-SGD. Notably, for the Rosenbrock function, ZO-RankSGD outperforms ZO-SGD, indicating superior robustness to additive noise. We speculate that this advantage stems from ZO-RankSGD relying solely on rank information for optimization, which may exhibit less variability under mild additive noise.
>
>    As one of the most important future directions, we hope to establish a well-defined formulation for the noisy ranking oracles and extend the theoretical analysis of ZO-RankSGD within this context.
>
>   ---
>
> ### 2. The second concern is "Narrow focus: the authors only focused on image generation with human feedback, which is also a bit limited ... it would be a great add if the authors can show that the method is also effective for tuning language models."
>
>    We agree with you that testing our algorithm for language models is certainly an interesting extension, and we did think about it for once. Unfortunately, since language models generally require much more computing power than diffusion models, and our computing resources are very limited, we were unable to perform such an experiment.
>
>    While saying that, we wish to argue that not including the experiment on LLM should not be overly criticized. On one hand, from our perspective, incorporating human feedback into improving diffusion models is more urgent for now as this area is still in its infancy, while there have been extensive works on language models. On the other hand, the main scope of this work is only to provide the first theoretical and principled framework for optimization with ranking oracles, and to inspire future research on incorporating this framework with more engineering consideration.
>
>    We sincerely hope the reviewer could understand our situation and take this explanation into the evaluation of the revised manuscript.

---

### Public Comment · ~Tongyang_Li1 · 2023-12-02

Dear authors,

We are deeply interested in your wonderful work. However, we encountered a couple of questions when trying to understand the technical details:

1. We are a bit confused about the proof of Corollary 1. Is it derived by taking the limit of m towards infinity, as in “Discussion on Lemma 4”? If so, in the limit where m becomes very large, would it be feasible to contemplate the (m, k)-ranking oracle model? Could the implementation of such a model potentially cause excessive costs?

2. Lemma 3 has an assumption of f satisfying \grad^{2} f(x) = c * I_d. This though sounds quite limited, as for general functions the Hessian matrix can be quite arbitrary. May I ask if there is any further justification for this assumption?

Your insights into these questions would be greatly appreciated!

---

### Meta-Review · Area_Chair_x3x5 · 2023-12-06

**Metareview:**

This paper studied zeroth order optimization problem with ranking oracle for feedback. The authors proposed ZO-RankSGD with a novel gradient estimator. The effectiveness of the algorithm is validated by theoretical guarantees of convergence and empirical results. The reviewers are unanimously positive about the paper, recognizing its novelty and wide range of applications.

**Justification For Why Not Higher Score:**

There are some remaining concerns such as empirical evaluations compared with preference-based RL/RLHF methods.

**Justification For Why Not Lower Score:**

The reviewers are unanimously positive about the paper and recognize its contribution.

---

### Decision · Program_Chairs · 2024-01-16

Accept (poster)